# Myco-Biosynthesis of Silver Nanoparticles, Optimization, Characterization, and In Silico Anticancer Activities by Molecular Docking Approach against Hepatic and Breast Cancer

**DOI:** 10.3390/biom14091170

**Published:** 2024-09-18

**Authors:** Noura El-Ahmady El-Naggar, Nada S. Shweqa, Hala M. Abdelmigid, Amal A. Alyamani, Naglaa Elshafey, Hoda M. Soliman, Yasmin M. Heikal

**Affiliations:** 1Department of Bioprocess Development, Genetic Engineering and Biotechnology Research Institute, City of Scientific Research and Technological Applications (SRTA-City), New Borg El Arab City 21934, Egypt; nelahmady@srtacity.sci.eg; 2Botany Department, Faculty of Science, Mansoura University, Mansoura 35516, Egypt; nadasalah2000@mans.edu.eg (N.S.S.); hudasoliman@mans.edu.eg (H.M.S.); 3Department of Biotechnology, College of Science, Taif University, Taif 21944, Saudi Arabia; h.majed@tu.edu.sa (H.M.A.); a.yamani@tu.edu.sa (A.A.A.); 4Botany and Microbiology Department, Faculty of Science, Arish University, Al-Arish 45511, Egypt; n_fathi@aru.edu.eg

**Keywords:** silver nanoparticles, biological synthesis, *Fusarium oxysporum*, face-centered central composite design, antitumor, molecular docking

## Abstract

This study explored the green synthesis of silver nanoparticles (AgNPs) using the extracellular filtrate of *Fusarium oxysporum* as a reducing agent and evaluated their antitumor potential through in vitro and in silico approaches. The biosynthesis of AgNPs was monitored by visual observation of the color change and confirmed by UV–Vis spectroscopy, revealing a characteristic peak at 418 nm. Scanning electron microscopy (SEM) and transmission electron microscopy (TEM) analyses showed spherical nanoparticles ranging from 6.53 to 21.84 nm in size, with stable colloidal behavior and a negative zeta potential of −15.5 mV. Selected area electron diffraction (SAED) confirmed the crystalline nature of the AgNPs, whereas energy-dispersive X-ray (EDX) indicated the presence of elemental silver at 34.35%. A face-centered central composite design (FCCD) was employed to optimize the biosynthesis process, yielding a maximum AgNPs yield of 96.77 µg/mL under the optimized conditions. The antitumor efficacy of AgNPs against MCF-7 and HepG2 cancer cell lines was assessed, with IC_50_ values of 35.4 µg/mL and 7.6 µg/mL, respectively. Molecular docking revealed interactions between Ag metal and key amino acids of BCL-2 (B-cell lymphoma-2) and FGF19 (fibroblast growth factor 19), consistent with in vitro data. These findings highlight the potential of biologically derived AgNPs as promising therapeutic agents for cancer treatment and demonstrate the utility of these methods for understanding the reaction mechanisms and optimizing nanomaterial synthesis.

## 1. Introduction

Nanotechnology encompasses the development, manipulation, and application of materials at the nanoscale, typically measuring less than one micron in size [1]. These nanoparticles demonstrate distinctive properties that are separate from their bulk counterparts, attributable to their size and other size-dependent characteristics [2]. Traditionally, there have been three main methods for nanoparticle synthesis: physical, chemical, and biological. However, a key challenge in advancing nanotechnology is developing new, efficient methods to produce nanoparticles with precise control over size, shape, and chemical composition while ensuring they are highly monodispersed [3].

Green nanotechnology offers a compelling alternative by harnessing biological resources like bacteria, fungi, yeast, plants, or their biomolecules (lipids and proteins) for an eco-friendlier approach to nanoparticle synthesis [4]. Silver (Ag) is a popular choice for nanoparticles due to its unique properties [5,6,7,8,9,10,11]. Other transition metals like gold (Au) and various other elements like platinum, iron (Fe), cadmium sulfide (CdS), and zinc oxide (ZnO) can also be used, each offering distinct characteristics and functionalities [12,13]. The use of Cu, Au, Pt, or Pd NPs for plasmonic applications is explored. AgNPs offer the best quality factor in plasmonic ability and exhibit the plasmonic band in an extensive range of wavelengths (from near ultraviolet to near infrared spectrum), unlike other NPs mentioned. Furthermore, Au and Pt NPs are almost 100 times more expensive than Ag NPs, but Cu NP utilization is significantly more problematic due to their high susceptibility to oxidation and the scarcity of recognized nanostructures [14,15].

The conventional processes for synthesizing nanoparticles through chemical and physical means often present inherent limitations, characterized by high costs and the utilization of hazardous chemicals, posing environmental and health hazards [16]. In contrast to chemical methods, biological methods for green synthesis provide a more sustainable and environment-friendly approach, highlighting their importance in advancing nanotechnology. Notably, nanoparticles can be synthesized through two primary approaches [17]: top-down (breaking down larger materials) and bottom-up (assembling individual atoms or molecules). Biological synthesis, akin to chemical methods but in contrast to physical processes, aligns with the bottom-up approach [17,18]. This sustainable approach exploits the enzymatic potential of fungi to facilitate the synthesis of metal nanoparticles, the formation of nanostructures, and the biomimetic mineralization process [19,20]. For instance, *F. oxysporum* and *Verticillium* species have been used to create magnetite nanoparticles (iron nanoparticles) using this method [21]. Some bioactive compounds from Fusarium species have been shown to effectively synthesize AgNPs, but little study has been conducted to determine which compound in these strains is specifically responsible for AgNPs synthesis [22].

Scaling up microbially derived nanoparticle production requires identifying optimal growth conditions. This approach leverages various optimization methods to achieve the most favorable conditions for large-scale synthesis. Build-Test-Fix (BTF) is a basic approach relying on available resources and trial-and-error [23]. One Factor at a Time (OFAT) focuses on changing one variable at a time, but it neglects interactions between factors and can lead to misinterpretations [24,25]. A more sophisticated approach is the Design of Experiments (DOE), which analyzes the effects of multiple controlled factors simultaneously and generates a prediction model, leading to more reliable optimization [26].

As a result of their exceptional chemical and physical characteristics, silver nanoparticles (AgNPs) have gained widespread popularity across various sectors, including medical, food, consumer goods, and industrial applications [3,27]. The broad range of applications for AgNPs can be primarily attributed to their remarkable qualities, which make them ideal candidates for a variety of purposes [28]. Notably, they function as potent broad-spectrum antibacterial agents, employed in various applications across industries, households, and healthcare facilities [28]. Additionally, AgNPs possess diverse applications across various fields, including medical device coatings, optical sensors, and cosmetic formulations. AgNPs are currently being investigated by the pharmaceutical and food sectors for their potential in drug delivery [29]. Moreover, ongoing research suggests their promise as anticancer agents, with the potential to augment the efficacy of existing cancer treatments [28]. AgNPs display pro-oxidative and pro-apoptotic properties, which make them valuable tools for implementation in photodynamic therapy against cancer cells [30,31,32]. Studies have confirmed their cytotoxic effects using the MTT assay, which measures mitochondrial activity in cells. These studies reveal variations in viability percentages across different cell lines treated with AgNPs [33]. Researchers have compared the effectiveness of AgNPs synthesized using algae with established anticancer drugs like cisplatin on MCF-7 cancer cell lines. These comparisons underscore the substantial anticancer potential of AgNPs [28].

Molecular docking is an advanced computational technique used in drug design for predicting the most effective and stable form of the ligand–receptor complex [34]. Thus, the application of new technologies can open the door to the discovery of novel medications with great therapeutic promise and represent a significant advancement in the treatment of disease [35,36]. Compound screening assays are among the molecular docking techniques that can support lead generation, validation, hit identification, and optimization procedures in addition to assessing the impact of the compounds on the therapeutic target. Due to technological advancements and the merging of computational science with biological and pharmaceutical research, methods like virtual screening are extensively utilized in drug design and discovery programs [37,38].

This investigation aimed to develop an optimized method for the green synthesis of AgNPs using the extracellular filtrate of *F. oxysporum*. To achieve this, we investigated the influence of four key process variables: incubation time, initial pH level, silver nitrate concentration, and temperature. We employed a design technique called face-centered central composite design (FCCD) to optimize these variables for efficient AgNPs biosynthesis. Following synthesis and optimization, the characteristics of the biosynthesized nanoparticles were analyzed. Finally, in vitro and in silico approaches via biosynthesized nanosilver antitumor activity against hepatic and breast cancer were evaluated through MTT assay and molecular docking computational technique.

## 2. Materials and Methods

### 2.1. Fungal Isolate and Growth Conditions

*Fusarium* sp. was procured from the Mycology Laboratory in the Department of Botany, Faculty of Science at Mansoura University in Egypt, and was utilized for the synthesis of AgNPs. To prepare the Potato Dextrose Agar (PDA) media, 200 g of potato extract, 20 g of dextrose, and 20 g of agar were dissolved in 1 L of distilled H_2_O. The solution was subjected to autoclaving (at 121 °C for 20 min). After cooling, the medium was thoroughly mixed and poured into Petri dishes. The obtained *Fusarium* sp. isolate was subsequently cultured on PDA medium and incubated at 25 °C for five days to achieve adequate growth of fungus. Following incubation, the culture was stored at 4 °C until further use in AgNPs biosynthesis.

### 2.2. Morphological and Molecular Identification of Fungal Isolate

#### 2.2.1. Morphological Identification: Light Microscopic Examination

The fungal isolate was examined using a digital light microscope (Optika Model B-380 Ver. 2, Italy) at magnifications of 100 and 400× to observe vegetative hyphae and reproductive structures (e.g., conidia and phialides). The observed morphological characteristics were compared with those of standard keys and reference materials for *Fusarium* sp. identification.

#### 2.2.2. Morphological Identification: Scanning Electron Microscopy (SEM) Analysis

The dried specimen was then mounted onto an SEM stub using the appropriate adhesive. Finally, the mounted sample was metal coated, typically with gold, using a sputter coater to enhance the electrical conductivity and improve the image quality [39]. SEM analysis was conducted using (JSM-6510 L.V, JEOL Ltd., Tokyo, Japan) at the Electron Microscope Unit, Mansoura University, Mansoura, Egypt.

#### 2.2.3. Molecular Identification: DNA Extraction, Polymerase Chain Reaction Amplification and Sequencing

The fungal isolate was cultivated on potato dextrose agar (PDA). Fresh mycelial samples were collected from three-day-old cultures. Approximately 10 mg of the harvested mycelia was dried using the freeze-drying method. The dried mycelia were then ground in liquid nitrogen using a mortar and pestle. The genomic DNA of the Fusarium sp. isolate was extracted using a modified CTAB protocol based on Panabieres et al. [40]. The purified DNA was dissolved in 50 µL TE buffer (containing 10 mM Tris-HCl and 0.1 mM EDTA, pH 8.0) for storage. All purified DNA preparations were stored at −80 °C for further use in PCR amplification.

Specific primers designed for amplifying the Internal Transcribed Spacer (ITS) region of rDNA were constructed using Primer3 (version 4.1.0) software [41] https://primer3.org/, accessed on 15 July 2023). These primers targeted conserved regions of the eukaryotic 18S rRNA gene: the forward primer (ITS_1_) with the sequence TCC GTA GGT GAA CCT GCG G and the reverse primer (ITS_4_) with the sequence TCC TCC GCT TAT TGA TAT GC. BIONEER Inc., Oakland, CA, USA) was used to supply the synthesized primers. ITS region was amplified using a 25 µL reaction volume for each sample in an Eppendorf Mastercycler. PCR program involved initial denaturation at 94 °C for 6 min, followed by 35 cycles of denaturation (94 °C, 45 s), annealing (56 °C, 45 s), and extension (72 °C, 45 s). A final extension at 72 °C for 5 min and a final hold at 4 °C completed the program. The successful amplification of the ITS region was verified by running the PCR products on a 2% agarose gel containing ethidium bromide (0.5 µg/mL) and visualizing the bands using a gel documentation system (MicroDoc system from Cleaver Scientific Ltd., Rugby, UK). This step validated the existence of the anticipated amplicons. Following this, the PCR products were kept at −20 °C for additional testing.

An ABI 3730XL DNA Analyzer (Applied Biosystems, Waltham, MA, USA) was used to sequence the PCR products and to determine their precise DNA sequences. This assembled sequence represented the fungal isolate’s 18S rRNA gene sequences. To identify the closely related sequences, we conducted a BLASTn search using the National Center for Biotechnology Information (NCBI) GenBank database (http://blast.ncbi.nlm.nih.gov/, accessed on 20 September 2023). The neighbor-joining method was performed for this analysis [42]. A phylogenetic tree was constructed using Mega 11 software [43] to illustrate the evolutionary relationships between the sequenced isolate and other fungal species based on their 18S rRNA gene sequences. The acquired sequence data were submitted to the NCBI GenBank database to attain accession numbers for future reference.

### 2.3. Extracellular Synthesis of AgNPs

#### 2.3.1. Fungal Biomass Preparation

The purified fungal biomass of *F. oxysporum* was acquired by cultivating the fungus on PDA media. Three fungal disks were added to 500 mL Erlenmeyer flasks containing 100 mL of fresh PDA broth. The biomass of the fungi was separated from the broth through filtration after the flasks were continuously shaken for 72 h at a temperature of 25 ± 1 °C. The fungal biomass was washed repeatedly with deionized sterile water to remove any residual medium. Then, 10 g (wet weight) was suspended in 100 mL sterile deionized water and shaken continuously (150 rpm) at 25 ± 1 °C for 24 h.

#### 2.3.2. Green Synthesis of AgNPs

In a 250 mL Erlenmeyer flask, 100 mL of the aqueous mycelial-free filtrate of *F. oxysporum* was treated with precisely weighed AgNO_3_ to reach a final concentration of 1 mM AgNO_3_ and incubated in an orbital shaker with continuous agitation (150 rpm) under dark conditions (to prevent the photooxidation of silver ions) for 72 h at 30 °C. The aqueous, mycelial-free filtrate without addition of AgNO_3_ was maintained as a control [44]. Following incubation, the mixture of AgNO_3_ and the aqueous mycelial-free filtrate of *F. oxysporum* turned dark brown; the visual color change indicates the formation of AgNPs. On contrast, no color change was noticed with the absence of AgNO_3_. The maximum absorbance of the green synthesized AgNPs was scanned in the wavelength of 300–600 nm range by UV–Vis spectroscopy.

### 2.4. Optimization of Biosynthesis Parameters Using Face-Centered Central Composite Design (FCCD)

This research investigated the influence of four variables on the synthesis of AgNPs using *F. oxysporum*. The variables under investigation were incubation time (h) (X_1_), initial pH level (X_2_), silver nitrate (AgNO_3_) concentration (mM/mL) (X_3_), and temperature (°C) (X_4_).

FCCD, a common statistical technique for sequential experimentation, was used to optimize these parameters. It offers several advantages, including reduced experimental costs and efficient information gathering. The FCCD design in this study comprised 30 trials, with each independent variable investigated at three levels: low (−1), middle (0), and high (+1). The center point, representing the middle level for all variables, was replicated six times. This replication serves two purposes: curvature evaluation and pure error estimation. All experiments were conducted twice to account for variability. The average amount of AgNPs obtained in each experiment was designated as the dependent variable (Y). Response surface regression analysis was applied to the experimental data obtained through the FCCD design to optimize the parameters for AgNPs synthesis. A second-order polynomial equation was employed to model the relationship between the independent variables (X_1_ − X_4_) and the response variable (Y). The equation is provided below:(1)$$Y=\beta_{0}+\sum_{i}\beta_{i}X_{i}+\sum_{ii}\beta_{ii}X_{i}^{2}+\sum_{ij}\beta_{ij}X_{i}X_{j}$$

$X_{i}$ are the coded levels of independent variables, which are coded as X_1_, X_2_, X_3_, and X_4_, and Y is the predicted response, $\beta_{ij}$ is the interaction coefficient, $\beta_{i}$ is the linear coefficient, $\beta_{ii}$ is the quadratic coefficient, and $\beta_{0}$ is the regression coefficient.

### 2.5. Biosynthesized AgNPs Characterization

A comprehensive characterization of the biosynthesized AgNPs is crucial to understanding their physicochemical properties, morphology, and stability. This section details the various analytical techniques employed to investigate these characteristics. These techniques include the following:

#### 2.5.1. Ultraviolet–Visible (UV–Vis) Spectroscopy

Atomic absorption spectroscopy (AAS) is used to determine the concentration of AgNPs; however, it does not reflect the concentrations of AgNP particles themselves. Instead, it reflects the concentration of Ag^+^ ions. Therefore, unlike the AAS spectrum, if UV–Vis spectra were used, the concentration of AgNPs would not include Ag^+^ ions but for only nanoparticles. Only AgNPs have UV–Vis spectra at wavelengths from 400 nm to 450 nm depending on their size and shape; however, the height of absorption intensity value will rely on the quantity of AgNPs in the solution. The concentration of AgNPs can be quantitatively determined based on the linear dependence between AgNPs concentration and UV–Vis absorbance up to 200 ppm [40,45]. In this study, AgNPs were prepared by using the aqueous mycelial-free filtrate of *F. oxysporum*, and the absorption spectra of the AgNPs solutions were measured between 300 and 600 nm, using deionized water as a reference (blank). To quantify AgNPs using the UV–Vis method, a calibration curve must be generated from the absorption intensity values of the UV–Vis spectrum determined from a known concentration solution. The variations in the UV–Vis spectra are indicative of the AgNPs’ particle size and the solution’s color. Different AgNPs concentration series (10–100 μg/mL) was obtained by diluting AgNPs stock solution (500 μg/mL) with deionized water. Those standards were measured together with samples to obtain the corresponding UV–Vis and the calibration curve for calculating sample concentrations.

#### 2.5.2. Scanning Electron Microscopy (SEM)

SEM analysis was employed to characterize the morphology (size and shape) of the biosynthesized AgNPs. Before the SEM analysis, the AgNPs suspension was centrifuged to separate the nanoparticles. A concentrated supernatant sample was then drop-cast onto a precleaned glass substrate. This allowed for complete desiccation, resulting in a uniform distribution of AgNPs on the surface. The dried samples were evaluated using a low-vacuum scanning electron microscope (JEOL JSM-6510 LV) operated at an accelerating voltage of 30 kV. This analysis was conducted at the EM Unit of Mansoura University, Egypt.

#### 2.5.3. Transmission Electron Microscopy (TEM)

TEM is a vital tool for examining the dimensions and shape of AgNPs with superior precision, encompassing many stages. Initially, a precise amount of the AgNPs suspension was deposited onto a carbon-coated copper grid (Type G 200, 3.05 μm diameter) procured from TAAP, USA. This grid was utilized for TEM analysis, and the sample was allowed to dry completely, ensuring an even distribution of AgNPs on the grid surface, and the JEOL JEM 2100, Tokyo, Japan apparatus operated at 200 kV. This analysis was conducted at the EM Unit of Mansoura University, Egypt. The sample preparation method was adapted from Wang [46].

#### 2.5.4. Energy-Dispersive X-ray Spectroscopy (EDX)

To verify the chemical composition of the synthesized AgNPs, EDX analysis was performed using an Oxford Instruments 6587 INCA X-ray microanalyzer on a JEOL JSM-5500 LV SEM, Tokyo, Japan operating at 20 kV. The EDX spectrum was acquired in spot profile mode, focusing on a region of the sample surface that is densely populated with AgNPs [44]. This approach enables the quantitative determination of the substances present in the analyzed sample, providing information about the presence and relative abundance of silver in the synthesized nanoparticles.

#### 2.5.5. Selected Area Diffraction Pattern (SAED)

SAED was employed to investigate the crystalline structure of AgNPs. This technique analyzes concentric diffraction patterns generated when a sample is bombarded with high-energy electrons. These diffraction rings correspond to crystal planes within the material. To perform SAED analysis, a very thin sample, typically approximately 100 nm thick, was required. In addition, high-energy electrons with an energy range of 100–400 keV were used. At these energies, the electrons behave like waves, enabling them to interact with the atomic structure of the sample and produce a diffraction pattern. By analyzing the arrangement of the diffraction rings in the SAED pattern, researchers can gain valuable insights into the crystal structure and lattice parameters of AgNPs.

#### 2.5.6. Zeta Potential Analysis

The stability of the silver nanoparticles (AgNPs) solution was assessed using a Zeta Potential Analyzer (Zetasizer ver. 7.01, Malvern Instruments, Westborough, MA, USA). Specifically, 5 mg of AgNPs was dispersed in 5 mL of ultrapure water and mixed for 30 min at room temperature. Subsequently, the zeta potential was determined at a 90° angle of detection at the Electron Microscope Unit of Mansoura University, Mansoura, Egypt.

#### 2.5.7. Fourier Transform Infrared Spectroscopy (FTIR)

FTIR spectroscopy was used to investigate the surface chemistry of the AgNPs produced. The functional groups associated with the nanoparticles’ surfaces were identified by analyzing the infrared absorption frequencies between 400 and 4000 cm⁻^1^ at a resolution of 4 cm⁻^1^. The sample preparation involved dispersing the AgNPs in a dry KBr matrix and compressing the mixture to form a clear disc. FTIR spectra were recorded using a JASCO FTIR instrument (Tokyo, Japan) with a KBr pellet as a background at the Spectroscopy Unit of the Faculty of Science, Mansoura University, Mansoura, Egypt.

### 2.6. Anti-Tumor Activity of Biosynthesized AgNPs

Cell lines and reagents: MCF-7 (breast) and HepG2 (liver) cancer cell lines (Nawah Scientific Company, El Mokattam, Egypt) were cultured in DMEM (Life Science Group L, UK) supplemented with 10% FBS (Cat No: S-001B-BR) and 1% penicillin/streptomycin (100 U/mL penicillin and 100 μg/mL streptomycin; Lonza, Basel, Switzerland). 3-(4,5-Dimethylthiazol-2-yl)-2,5-diphenyltetrazolium bromide; MTT; Serva, Heidelberg, Germany) with DMSO (Sigma-Aldrich, St. Louis, MO, USA, Cat. No. 20385.02) was also employed to assess the antitumor potential of biosynthesized AgNPs from *F. oxysporum*.

MTT Assay: MTT (3-[4,5-dimethylthiazol-2-yl]-2,5 diphenyl tetrazolium bromide) assay was used to gauge the cytotoxicity and proliferation of cancer cells treated with biosynthesized AgNPs. MCF-7 and HepG2 cells (5 × 10^3^ cells/well in 100 μL) were seeded in 96-well plates and incubated for 24 h at 37 °C, 5% CO_2_ for adherence in a humidified incubator. Following adherence, MCF-7 and HepG2 cells were cultured in single wells for each concentration: control (no drug) and a single concentration of drug (50 μM) solubilized in 10 mM DMSO for 48 h. Cell growth inhibition was measured at 570 nm (BioTek, Winooski, VT, USA, Elx800). Cell viability (%) was calculated using the following equation:
Viability (%) = (Test OD/Control OD) × 100(2)
OD, optical density, and cytotoxicity (%) were then determined as 100—viability (%). For concentrations with <40% viability, cells were cultured in duplicates for a serial dilution (2.7–5.4 μg/mL) to calculate the IC_50_ using GraphPad Prism 9 (GraphPad Software, Inc., La Jolla, CA, USA).

### 2.7. In Silico Anticancer Activity by Molecular Docking Approach

We utilized molecular docking analysis to explore how the manufactured AgNPs interacted with the FGF19 protein, a crucial element in the growth and survival of most hepatocellular carcinomas, and the BCL-2 protein, a key player in the regulation of breast cancer apoptosis. We obtained the structures of these proteins from the Protein Data Bank (PDB IDs: 2W3L and 6YI8, respectively). After eliminating ligands and water, we added polar hydrogen and Kollman charges to the protein structures to prepare them for docking. To construct the structure of a single silver atom, we used Avogadro 1.2.0 and applied the UFF force field to minimize energy by adding hydrogen atoms, distributing partial charges, and reducing energy [47,48].

### 2.8. Statistical Analysis and Modeling

The statistical software Design-Expert^®^ (Version 12.0.0, Stat-Ease, Inc., Minneapolis, MN, USA) was used to design the experiment and analyze the resulting data. A multiple regression analysis was conducted on the experimental data to achieve the following.

#### 2.8.1. Analysis of Variance (ANOVA)

ANOVA was conducted to determine the statistical significance of the factors and their interactions with the response variable. This analysis assesses the impact of each factor on the observed variations in the response.

#### 2.8.2. The *p*-Value, F-Value, and Confidence Levels

These statistical parameters were calculated to assess the importance of the model and the individual factors. *p*-values indicate the likelihood of observing results by chance alone. F-values measure the variance explained by the model compared with the residual variance. The confidence levels represent the range within which the true mean is likely to be.

#### 2.8.3. Determination Coefficient (R^2^) and Adjusted R^2^

R^2^ reflects the fraction of variance in the response variable as explained by the fitted model. The adjusted R^2^ penalizes the model for additional terms, providing a more reliable assessment of fit, especially when comparing models with different numbers of predictors.

STATISTICA^®^ software (Version 8, StatSoft, Inc., Tulsa, OK, USA) was used to generate three-dimensional (3D) surface plots. These visual representations provide insights into the relationships between the independent variables and response variables, allowing for a better understanding of the optimization process.

#### 2.8.4. The Autodocking

4.2 program carried out the docking using 50 genetic algorithm runs, 500 population sizes, and 25 million evaluations. Following the docking process, Discovery Studio 2021 and Chimera evaluated the interactions between the AgNPs and amino acid residues [49].

## 3. Results and Discussion

This investigation presented a biological approach for nanoparticle biosynthesis utilizing *F. oxysporum*, aiming to achieve precise size control, a critical factor in nanotechnology. This eco-friendly method offers distinct advantages over traditional physical and chemical techniques. It eliminates the need for expensive equipment and processes, making it cost-effective. Additionally, it avoided the use of hazardous chemicals, promoting a more environmentally friendly approach. Furthermore, the biocompatible nature of this method potentially reduces risks associated with nanoparticle applications [50]. The selection of fungi, particularly *F. oxysporum*, is attributed to their abundant biomass availability and significant production of extracellular enzymes. These enzymes not only facilitate subsequent processing but also potentially enhance nanoparticle yields compared with bacteria [51]. Significantly, fungi not only facilitate rapid and eco-friendly sustainable biosynthesis, but they also promote a crucial property: good size uniformity (mono-dispersity) of the nanoparticles [52].

Rai, et al. [53] reported that the cytotoxicity of nanoparticles from *Fusarium* species, in particular AgNPs, and their underlying molecular mechanisms have been demonstrated in numerous studies that have applied a variety of model cell lines. The biogenic nanoparticles are capped with a corona, which is composed of natural molecules, including proteins. This nanoparticle corona has a substantial impact on the biological response. The corona can be classified into two types: hard corona and soft corona, based on the surface affinity as well as exchange rate. While the rigid coronas are rigid for entry into the cellular system, the soft corona proteins function as “vehicles” for the silver ions. Several reports demonstrate that the nanoparticle corona’s proteins interact with the cells rather than the nanoparticles themselves. For this reason, the corona formation and composition have major effects for both internalization and toxicity [54]. In addition to the protein charges, these functional groups also modulate the cytotoxic properties of the nanoparticle corona. The biological activity of nanoparticles is significantly influenced by their surface charge. In summary, biological fluid properties, composition, size, shape, and surface characteristics of particles all influence the corona structure and, thus, their detrimental effects on the environment and the health of humans [53].

### 3.1. Fungal Identification

The fungal isolate utilized for AgNPs biosynthesis in this study was identified as *F. oxysporum.* Confirmation involved the application of both morphological and molecular methods. Morphologically, the fungus presented characteristic pinkish-white aerial hyphae on plates, alongside the presence of macroconidia, microconidia, and chlamydospores upon microscopic examination (Figure 1A–C). Through scanning electron microscopy (SEM) analysis, a highly detailed visualization of the growth pattern of *F. oxysporum* was confidently observed, as illustrated in Figure 1D. The robust validation of the morphological identification was provided through molecular confirmation, ensuring reliable verification. To achieve this, the ITS gene region, a highly conserved region within fungal genomes, was amplified from the isolated fungal DNA. Following amplification and purification, the PCR product was sequenced, revealing a unique nucleotide sequence.

This sequence served as a molecular fingerprint for the isolate and was subsequently compared against the comprehensive GenBank database using the Basic Local Alignment Search Tool (BLAST). Phylogenetic analysis (Figure 2) of the obtained ITS sequence revealed 100% similarity to known *F. oxysporum* sequences, conclusively supporting the initial morphological identification of the fungus as *F. oxysporum*.

### 3.2. Biosynthesis and Characterization of Silver Nanoparticles (AgNPs)

The production of AgNPs by *F. oxysporum* was verified by observing a brown color in the culture medium. The observed color change served as a well-established indicator of nanoparticle formation [55]. To further characterize the optical properties of the biosynthesized AgNPs, UV–Vis spectrophotometry was employed within the range of 300–600 nm (Figure 3). The analysis of the optical absorption spectra is a powerful tool for characterizing metal nanoparticles. These spectra are influenced by a phenomenon proven as surface plasmon resonance (SPR). SPR specifically affects the position of the absorption peak, with the peak wavelength corresponding to the size of the nanoparticles [56]. According to the Mie theory [57], spherical nanoparticles typically exhibit a single SPR band in their spectra. By contrast, particles with more complex shapes (anisotropic) may exhibit multiple SPR bands depending on their specific geometry [58]. As the symmetry of the nanoparticles decreases, the number of detectable SPR peaks increases. Additionally, the SPR peak position in aqueous solutions shifts towards longer wavelengths with increasing particle size [59]. Other factors influencing the position and shape of the SPR peak include the particle size, stabilizing molecules adsorbed on the surface, and the dielectric constant of the surrounding medium [59].

In this study, analysis of the UV–Vis spectrum revealed a single peak for the biosynthesized AgNPs, with the highest absorbance at 418 nm, which typically indicated the characteristic surface plasmon resonance (SPR) of AgNPs (Figure 3A). This SPR peak, arising from light-induced electron oscillation on the AgNPs surface, offers insights into their size and morphology. This peak position suggested the creation of spherical NPs, a finding that was subsequently verified through TEM analysis. Our results aligned well with El-Ahmady El-Naggar et al. [11], who reported a similar Surface Plasmon Resonance (SPR) peak at 425 nm for AgNPs synthesized using the phycocyanin pigment. This observation reinforced the established principle that the SPR peak in metal nanoparticles shifts towards longer wavelengths as the particle size increases [56]. However, it is important to note that variations in SPR peak positions can occur because of several factors besides the particle size. Ahmad et al. [60] and El-Naggar et al. [9] reported SPR peaks at 415 and 430 nm, respectively, for AgNPs synthesized using different methods. These variations can be attributed to factors such as the dielectric constant of the surrounding medium and the presence of stabilizing molecules on the nanoparticle surface [59]. On the other hand, the specific SPR spectra of the biosynthesized AgNPs obtained from the culture supernatant of *Streptomyces viridodiastaticus* SSHH-1, *Streptomyces viridochromogenes*, and *Streptomyces aegyptia* NEAE 102 exhibited a prominent peak at 400 nm, indicating the presence of AgNPs. The peak intensity was observed to increase as the reaction progressed [5,6]. While the precise mechanism for the synthesis of nanoparticles remains to be definitively confirmed, it is understood that the process involves an enzyme known as NADH-dependent nitrate reductase [10].

### 3.3. Optimization of Silver Nanoparticle Biosynthesis

A face-centered central composite design (FCCD) was exploited to improve the biogenesis of AgNPs using *F. oxysporum*. Four key variables (pH, incubation time, silver nitrate concentration, and temperature) were investigated at three levels each (−1, 0, and +1), as shown in Table 1 and Figure 3B. A total of 30 experimental runs were performed, incorporating center point replicates (runs 1, 2, 10, 13, 27, and 28) for increased data reliability. AgNPs yield varied significantly across experiments, ranging from 35.45 µg/mL (run 16) to 97.32 µg/mL (run 9). The lowest yield occurred under conditions of pH 7, 72 h incubation, 0.5 mM/mL silver nitrate, and 30 °C (run 16). Conversely, the optimal conditions for AgNP production were identified at pH 7, 72 h incubation, 1.5 mM/mL silver nitrate concentration, and 40 °C (run 9). These conditions resulted in the highest yield of 97.32 µg/mL. Table 1 showed the actual and predicted AgNPs yields for all experimental runs. The observed values generally align with the predicted values, indicating the validity of the employed FCCD model.

### 3.4. Model Fit and Statistical Analysis

The fit of the model was evaluated using the coefficient of determination (R^2^) from the ANOVA analysis (Table 2). R^2^ quantifies the proportion of response variance explained by the experimental variables and their interactions [5]. The model exhibited a high degree of fit, with a determination coefficient (R^2^) of 0.9952. R^2^ values are ranged from 0 to 1. A value close to 1 indicates that the model’s predictions aligned well with the observed data, reflecting a better fit [11,61]. Furthermore, the adjusted R^2^ (0.9907) and predicted R^2^ (0.9745) were also exceptionally high (Table 2). Adjusted R^2^ accounts for the impact of model complexity on the explained variance, while predicted R^2^ estimates the model’s ability to predict responses for new experiments [62]. This remarkable concordance between adjusted R^2^ and predicted R^2^ (falling within the 20% agreement threshold proposed by El-Naggar and Rabei [63] substantiated the model’s exceptional predictive power, thereby highlighting its potential for practical applications.

To assess the significance of the model’s factors and their interactions, *p*-values were calculated. In general, a smaller *p*-value indicates a stronger association between the factor and the response variable [8]. The analysis of variance (ANOVA) revealed a highly significant model (F-value = 220.94, *p*-value < 0.0001), indicating that at least one of the studied factors (incubation time (X_1_), initial pH level (X_2_), AgNO_3_ concentration (X_3_), and temperature (X_4_) significantly affected the rate of AgNPs biosynthesis. This was further supported by the statistically significant *p*-values (<0.05) for the linear coefficients of all four factors. Additionally, their high F-values (X_1_: 337.02, X_2_: 136.08, X_3_: 1072.17, X_4_: 69.31) suggest a strong influence on the response variable. These findings imply that slight variations in incubation time, initial pH, AgNO_3_ concentration, and temperature can significantly impact the rate of biosynthesis. Statistical analysis revealed a strong dominance of linear relationships between the process variables (incubation time, initial pH, AgNO_3_ concentration, and temperature) and the rate of biosynthesis. This is evident by the *p*-values (<0.05) for the linear coefficients of all four factors.

Conversely, the quadratic terms (accounting for potential curvature) were non-significant, as shown by their high *p*-values (0.7117–0.9281) and low F-values (0.01–0.14). These findings implied that, within the studied range, the impact of each factor on the response can be best described by a straight line, with minimal to no curvature observed. The analysis of the coefficients’ *p*-values revealed the significance of the interactions among the four factors under examination. Specifically, it was determined that several factors significantly affected the results (*p*-value < 0.0001). These included how long the solution was incubated (incubation time), how acidic it was at the start (initial pH), and the temperature, all in combination with the concentration of AgNO_3_. The relationships between incubation time, the initial pH level, and temperature were non-significant.

The analysis of coefficient signs, whether positive or negative, has provided valuable insights into the direction of the effects, whether they enhance or inhibit, and the interactions between process variables [64]. Notably, two-factor interactions may exhibit opposing (negative coefficient) or complementary (positive coefficient) effects on the response variable [65]. As shown in Table 2, it is evident that all linear coefficients, including incubation time (X_1_) and AgNO_3_ concentration (X_3_), displayed positive values, indicating a positive influence on AgNPs production. Similarly, all significant interaction terms showed positive coefficients, suggesting a synergistic effect. In simpler terms, the combined effect of these interacting factors on biosynthesis surpassed the sum of their individual impacts. It is important to note that all interactions among the four variables (*p*-value < 0.0001) greatly enhanced AgNPs biosynthesis, underscoring the critical role of these interactions in optimizing the production process [64,65]. An adequate precision value serves as a measure of the signal-to-noise ratio in the model, with values greater than four indicating good model predictability [66]. Based on the results obtained by the model with a precision of 71.71, it is evident that the model is suitable for examining the design space, which aligned with the desirable range. This suggested that the model can effectively distinguish between real effects and background noise, making it a reliable tool for further investigation.

To determine the optimal model for describing the biogenesis of AgNPs employing *F. oxysporum*, a comparison was made between linear, two-factor interaction (2FI), and quadratic models based on fit summaries. The selection of the most suitable model was dependent on two crucial factors: (1) statistically significant model terms and (2) non-significant lack-of-fit tests. The analysis of the FCCD fit results (Table 3) revealed that the 2FI model was the most appropriate. This model exhibited a highly significant level (*p*-value < 0.0001), indicating a strong correlation between the model’s predictions and the actual observations. In contrast, the lack-of-fit test for the quadratic model showed no concerns (F-value = 1.00, *p*-value = 0.5327) (Table 3), suggesting that the model adequately fits the data without any unexplained variations. Although the quadratic model had the lowest standard deviation (1.23), it also had the highest R-squared value (0.9952).

A second-order polynomial equation was employed to model the relationship between various factors influencing AgNPs biosynthesis using *F. oxysporum*. These factors, represented as independent variables (X_1_, X_2_, X_3_, and X_4_), included incubation time (X_1_), initial pH level (X_2_), AgNO_3_ concentration (X_3_), and temperature (X_4_). These variable settings were analyzed using a statistical method called multiple regression. The resulting equation allows for the prediction of the dependent variable (Y), which represents the final AgNPs biosynthesis yield, based on these variable settings:Y = 9.73 + 0.14 X_1_ − 0.32 X_2_ + 2.14 X_3_ − 0.25 X_4_ + 0.24 X_1_X_2_ − 0.68 X_1_X_3_ + 0.67 X_1_X_4_ − 0.29 X_2_X_3_ + 0.22 X_2_X_4_ − 0.26 X_3_X_4_ − 0.81 X_1_^2^ − 0.56 X_2_^2^ − 2.36 X_3_^2^ − 0.66 X_4_^2^(3)

The given equation predicts AgNPs yield (Y) considering key factors: X_1_ (incubation time), X_2_ (initial pH), X_3_ (silver nitrate concentration), and X_4_ (reaction temperature).

### 3.5. Three-Dimensional Surface Plots Reveal Optimal Conditions and Interactions

Special 3D graphs (response surface graphs) to identify the best conditions (variable levels) for maximizing AgNPs production. These graphs also revealed how the different factors (initial pH, incubation time, concentration of AgNO_3_, and temperature) interact with each other to influence AgNPs production (Figure 4). According to Kamat, et al. [67] temperature, time, and reactant concentration are crucial in determining the number of yielded particles, their distribution, and size. To find out the optimal conditions for maximum AgNPs biogenesis by *F. oxysporum,* the amount of AgNPs produced (mg/mL) was shown on the Z-axis of the graph, while two other factors were fixed at the center (zero) levels. 

Figure 4A–C depicts how incubation time interacts with other factors to influence AgNPs biosynthesis. The plots show a consistent rise in AgNPs production with longer incubation periods, reaching a peak around 72 h. Deviations from this optimal time (shorter or longer) resulted in decreased yield. This highlights the importance of finding the ideal incubation time for maximizing AgNPs production.

Figure 4A,D,E illustrated the interactions between the initial pH and other key factors (incubation time, silver concentration, and temperature), influencing the biosynthesis of AgNPs. These plots reveal that an initial rise in pH enhanced the production of AgNPs, reaching its peak at approximately pH 7. However, further changes in pH lead to a decline in AgNPs. It is commonly acknowledged that pH plays a crucial role in controlling the final size, shape, and stability of AgNPs [68]. Singh et al. [69] proved the synthesis of spherical AgNPs with diameters of 12–17 nm using *Hibiscus* leaf extract. 

Figure 4F highlighted the influence of temperature and silver nitrate concentration. As silver concentration increases towards an optimal level (around 1.5 mM/mL), AgNPs biosynthesis rises, reaching a maximum of approximately 97.32 mg/mL. This trend reflects the importance of the silver precursor amount. However, exceeding the optimal concentration leads to a decrease in AgNPs yield.

### 3.6. Evaluating Model Adequacy

A key step in evaluating our model is to verify that the errors (residuals) follow a normal distribution. A valuable tool for this purpose is the normal probability plot (NPP), shown in Figure 5. The near-straight line in Figure 5 suggested that the residuals are indeed normally distributed [9]. The residuals represented the difference between the experimentally obtained AgNPs biosynthesis values and the amounts the model predicted. Smaller residual values signified a closer match between the model predictions and experimental results.

In addition to NPP, Figure 5 also presented the Box-Cox plot produced during the transformation of the model for AgNPs biogenesis (Figure 5D). The green line depicted a λ value of 0.7 as optimal, whereas the blue line represented the present value of lambda (λ = 1). The red lines indicated the 95% confidence interval, encompassing the range of values within which the true value is likely to fall with 95% certainty (0.34 and 1.02, respectively). As the blue line (current lambda) lies between the two red vertical lines, the model is considered to be in the optimal zone [11]. These results prove that the model accurately reflects the observed behavior in the experiments without requiring data transformation. 

Furthermore, Figure 5B displays a plot of the residuals versus the model’s forecasts for the amount of AgNPs produced. The graph clearly illustrated that the residuals are randomly dotted around the zero line. The residuals exhibited a homogenous distribution around zero, further reinforcing the normality observed in NPP. Additionally, Figure 5C depicted a scatter plot illustrating the predicted AgNPs biosynthesis values against the corresponding experimentally determined actual biosynthesis values. The diagonally aligned data points indicated that the model’s predictions are in close agreement with the actual outcomes, thereby demonstrating its high level of accuracy in the conducted experiments [11]. 

### 3.7. Desirability Function and Optimal Conditions

The desirability function was utilized to determine the best settings for maximizing AgNPs biosynthesis. This function assigns values ranging from 0 (undesirable) to 1 (desirable) to different response levels [66]. While desirability values are often calculated mathematically before experimental validation [70], we used the model’s predictions for the actual variables tested: incubation time (72 h), initial pH (7), AgNO_3_ concentration (1.5 mM/mL), and temperature (40 °C). Based on these factors, the model predicted a maximum AgNPs biosynthesis value of 96.77 µg/mL (Figure 6).

### 3.8. Characterization of Biosynthesized AgNPs

Several techniques were utilized to describe the morphologic and physical properties of the AgNPs synthesized by *F. oxysporum*, including UV–VIS spectroscopy, SEM, TEM, EDX, zeta potential, and FTIR.

#### 3.8.1. Morphology and Size Analysis

The SEM analysis results demonstrated the morphology of AgNPs, as shown in Figure 7A. The micrograph principally exhibited many spherical nanoparticles with a consistent and uniform distribution, with no noticeable agglomeration. The size of individual nanoparticles varied from 1 to 50 nm, with only a few isolated instances of agglomeration [71,72].

Employing TEM provided a deeper understanding of the material’s morphology and size distribution of AgNPs (Figure 7B–D). The image depicted both individual and aggregated particles with a size range of 6.53–21.84 nanometers. Notably, the presence of a capping agent surrounding the protein-coated nanoparticles contributed to their separation even within the aggregates. This capping agent generates a repulsive force between particles that prevents excessive aggregation. TEM micrographs primarily revealed spherical AgNPs, although some variations in shape and size were observed, consistent with other biological systems [73].

Selected area electron diffraction (SAED) analysis, a crystallographic technique performed within TEM, was employed to confirm the crystalline nature of the biosynthesized AgNPs. Analyzing a single drop from the nanocolloidal solution using SAED revealed a diffraction pattern for the nanosilver appearing as bright spots on a dark field (Figure 7E). This pattern reflected the crystalline structure of the nanosilver by displaying characteristic diffraction rings. The presence of these rings in the SAED micrograph confirmed the successful synthesis of crystalline AgNPs [74].

#### 3.8.2. Elemental Composition and Surface Charge Analysis

Energy-dispersive X-ray spectroscopy (EDX) analysis performed within the TEM (Figure 7D) provided insights into the elemental composition of the AgNPs. The EDX profile revealed a strong silver peak corresponding to 34.35 wt% of the sample. Additionally, weak peaks were observed for copper (0.66 wt%), calcium (0.32 wt%), and chlorine (11.12 wt%) (Figure 7F). The presence of these minor elements could potentially originate from biomolecules attached to the AgNPs surface [75].

Zeta potential analysis was conducted to evaluate the surface charge of the biosynthesized AgNPs (Figure 8A). The measured value was −15.5 mV with a single peak, indicating the presence of repulsive forces among the nanoparticles. A high negative or positive zeta potential value signifies strong repulsion between particles in suspension, preventing them from aggregating. In contrast, low zeta potential values indicate weak repulsion between the particles. This weak force enables them to approach one another and aggregate, which could potentially result in particle flocculation [76]. In this case, the −15.5 mV zeta potential indicated moderate stability of the AgNPs suspension.

#### 3.8.3. Functional Group Identification Using FTIR

Fourier-transform infrared spectroscopy (FTIR) analysis was employed to detect the functional groups responsible for stabilizing and reducing the AgNPs (Figure 8B). The FTIR spectrum of the *F. oxysporum*-synthesized AgNPs displayed vibrational frequencies at 3448 cm⁻^1^, 2927 cm⁻^1^, 1651 cm⁻^1^, 1243 cm⁻^1^, 994 cm⁻^1^, and 509 cm⁻^1^. The band observed that 1651 cm⁻^1^ corresponds to the bending vibrations of the chemical bonds within the protein’s building blocks, called amides (I and II) [77]. Additionally, the presence of primary amines was indicated by the stretching vibration peak at 3448 cm⁻^1^ [55]. The data obtained from the FTIR analysis indicated that phenols and proteins likely served a dual role. They functioned as reducing agents, converting silver ions to silver radicals, and simultaneously functioned as capping and stabilizing agents for the synthesized AgNPs [74].

### 3.9. Evaluation of Antitumor Activity

Previous studies have reported the antitumor potential of AgNPs and Ag^+^ ions [78,79]. For instance, the cell inhibition against human breast cancer cells (MDA-MB-231) exhibited by AgNPs produced using an extract from the *Acalypha indica* plant was 40% [59]. In this study, we investigated the in vitro antitumor activity of biosynthesized AgNPs from *F. oxysporum* against two common types of cancer: liver (HepG-2) and breast (MCF-7) cancer cells (Table 4, Figure 9 and Figure 10). The MTT assay enabled the detection of the quantity of AgNPs required to reduce cell growth by 50% (IC_50_), as shown in Figure 9 and Figure 10. The results revealed potent antitumor activity with IC_50_ values of 7.6 µg/mL against HepG-2 cells and 35.4 µg/mL against MCF-7 cells (Table 4).

The enhanced cytotoxicity of AgNPs might be attributed to their small size, facilitating penetration into tumor cells [80]. Several studies propose that AgNPs could interact with enzymes rich in a specific chemical group (thiol group) containing sulfur, leading to disruptions in cellular protein function and subsequent changes in cellular chemistry [81]. It is essential to consider that the toxicity of silver nanoparticles is contingent upon various factors, including concentration, size, and shape. In the context of green synthesis methods, these factors are subject to the chemical composition of the fungal extracts employed, potentially resulting in divergences in the biological efficacy of the resultant AgNPs [81,82].

#### Molecular Docking Approach

Many biological functions, including antioxidant enzymes, anti-metastasis, and anti-biofilm activities, are inhibited when metal ions, like silver, interact with different biomolecular targets [83]. This study found that silver nanoparticles interact with the amino acid residues of BCL-2 and FGF19 proteins through metallic bonding. We used the Autodock parameter version 4.2 in the molecular docking process to investigate the biological interactions between the silver nanoparticles and protein targets, such as FGF19 and BCL-2 proteins. Figure 11 depicts the binding of silver nanoparticles with the BCL-2 and FGF19 proteins during the docking approach. During the docking process, the silver atom’s binding energy was less than 0 kcal/mol, indicating that it was more likely to interact with the binding residues of the target proteins at its free energy, as shown in Table 5. During the docking process, we formed 50 docking poses and directed all 500 instances towards a single binding site in each protein, demonstrating the strong attraction of silver atoms to these sites. The interaction analysis showed that the constructed silver atoms can interact with the LEU96, ASP99 residues of BCL-2, CYS552, THR499, and ALA501 residues of FGF19 through the metallic bond, as shown in Figure 11 and Table 5. This is similar to a study conducted by Buglak et al. [84], which found that cysteine, aspartic acid, and tyrosine had the highest binding energy values with silver ions. In addition, the activity of BCL-2 and FGF19 proteins can be inhibited by the interaction of Ag nanoparticles with alanine, cystine, leucine, and threonine amino acid residues, according to Akram et al. [49].

The BCL-2 protein family members play important roles in controlling apoptotic cell death. Abnormal overexpression of pro-survival BCL-2 family members or abnormal reduction of pro-apoptotic BCL-2 family proteins, both resulting in the inhibition of apoptosis, are frequently detected in diverse malignancies [85]. Therefore, BCL-2 family members and their regulators are attractive targets for the development of anticancer therapeutics [85]. One of the characteristics of cancer is evasion of apoptosis [86]. The counterbalance between pro-apoptotic and anti-apoptotic of the BCL-2 (B-cell lymphoma-2) family has a substantial role in the apoptosis process [87]. Extensive studies suggest that the upregulation of the anti-apoptotic BCL-2 protein is commonly observed in numerous cancer cells. This overexpression is the primary reason why neoplasm cells are able to evade apoptosis and develop resistance to chemotherapy [88]. Thus, the control of the anti-apoptotic BCL-2 protein plays a crucial role in the treatment of cancer. Several inhibitors targeting the BCL-2 protein have been created, including obatoclax [89], venetoclax [90], and navitoclax [88].

On the other hand, the growth factors of the FGF family perform several roles, such as controlling metabolism, cell division, proliferation, and survival. While some research has revealed that FGF19 is also implicated in hepatoblastoma [91] and cholangiocarcinoma [92]. Out of the 353 patient samples in the Cancer Genome Atlas (TCGA) database, around 7% (25 individuals) exhibit a high copy number of FGF19 in hepatocellular carcinoma (HCC). Although no significant difference was observed between the two cohorts or in the overall survival (OS) between those with high and low FGF19 expression, patients with higher expression of FGF19 in HCC had shorter median survival when compared with a low-expression cohort (http://kmplot.com/analysis, last accessed 2 April 2021). FGF19 signaling has a role in the development and progression of hepatocellular carcinoma (HCC) by influencing both direct and indirect signaling pathways that are shared with other factors [93].

## 4. Conclusions

Nanoparticles have promising applications in the treatment of a variety of disorders, particularly cancer treatment. In this study, AgNPs were successfully synthesized using the eco-friendly fungus *F. oxysporum*. The key factors influencing the AgNPs yield were incubation time, initial pH, AgNO_3_, and temperature. A mathematical model was established to predict the optimal biosynthesis conditions, achieving a yield of 97.32 µg/mL. Characterization confirmed the spherical morphology and crystalline nature of the biosynthesized AgNPs, with a size range of 6.53–21.84 nm. Finally, the in vitro results revealed that AgNPs have potent antitumor activity against HepG2 cells and MCF-7 cells with IC_50_ values of 7.6 µg/mL and 35.4 µg/mL, respectively. In silico data from computational docking of FGF19 and BCL-2 proteins showed that they play significant roles in hepatocellular and breast cancer development, respectively. Given this condition, functionally blocking FGF19 and BCL-2 can restore tumor cell apoptosis. Overall, this study demonstrated the potential of biogenic AgNPs for cancer treatment, while FGF19 and BCL-2 family chemicals showed great promise as tumor therapeutic targets or indicators of tumor disorders, providing hope for targeted tumor therapy. Continued study in this area should aim to characterize nanosilver’s influence on cellular and molecular targets that control apoptosis, as well as to investigate its potential for clinical application. Indeed, the researchers have improved the biocompatibility of conventional nanomaterials by modifying their surface and optimizing their composition in several ways.

## Figures and Tables

**Figure 1 biomolecules-14-01170-f001:**
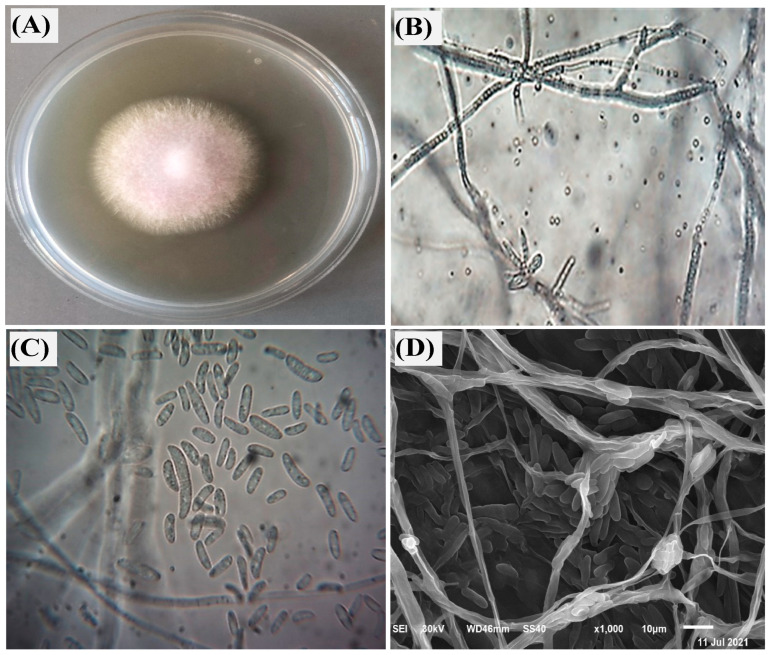
Morphological and structural identification of *Fusarium oxysporum,* where (**A**) characteristic growth of *F. oxysporum* on PDA medium after 7 days incubation at 25 °C; (**B**,**C**) microscopic characteristic showing septate mycelium, macroconidia, and microconidia on light microscope at 100 and 400×, respectively; and (**D**) SEM image at 10 µm.

**Figure 2 biomolecules-14-01170-f002:**
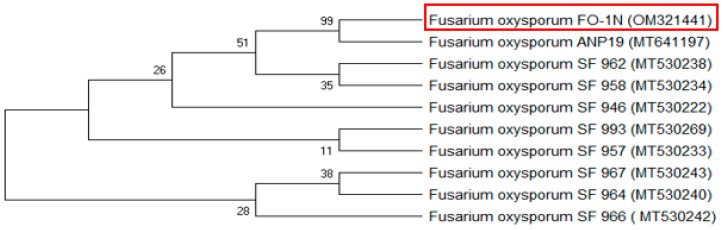
Phylogenetic tree generated from *F. oxysporum* isolates sequences of internal transcribed spacer ITS regions at bootstrap values of 1000 replicates. Abbreviations in brackets indicate accession numbers and the red square represents the studied isolate.

**Figure 3 biomolecules-14-01170-f003:**
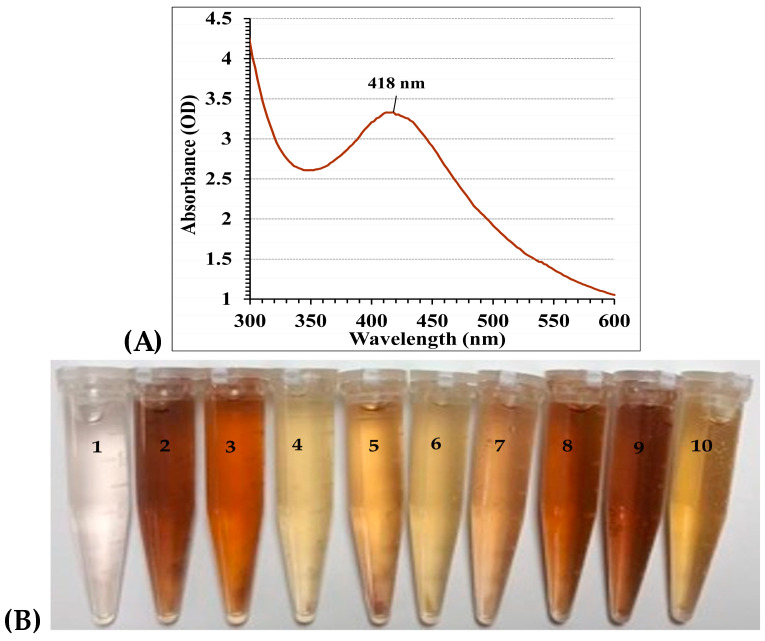
(**A**) UV–Vis spectral scan (from 300–600 nm) of nanosilver (**B**) Change in color depends on the trial conditions: (1) Filtrate without AgNO_3_ (control), (2) Run 3; (3) Run 4; (4) Run 6; (5) Run 7; (6) Run 12; (7) Run 14; (8) Run 19; (9) Run 23; and (10) Run 25. All After incubation time 24 h.

**Figure 4 biomolecules-14-01170-f004:**
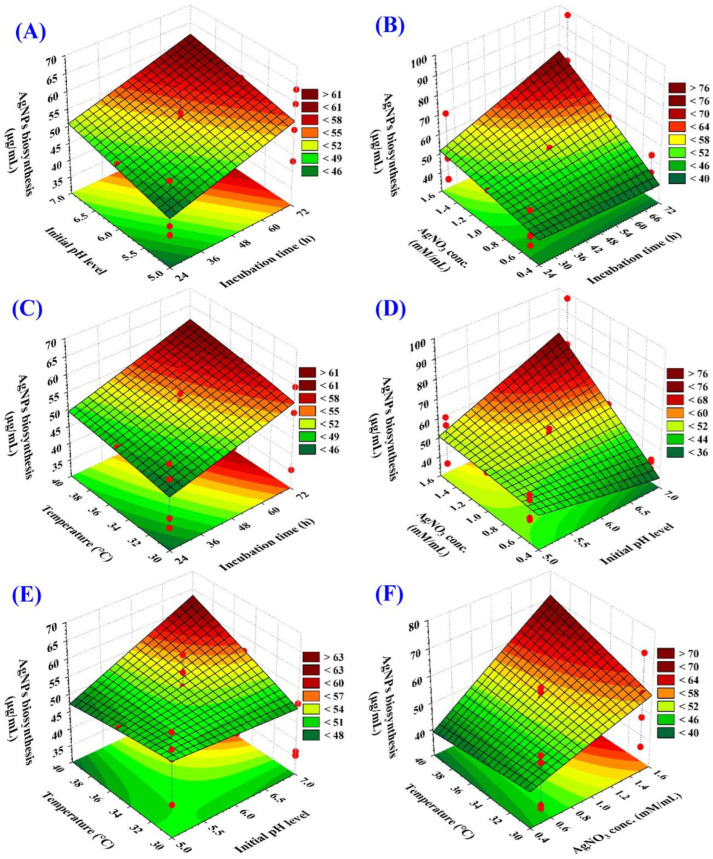
Subfigures (**A**–**F**) showing the three-dimensional plots of the mutual effects of incubation time (X_1_), initial pH level (X_2_), AgNO_3_ conc. (X_3_), and temperature (X_4_) on silver nanoparticle biosynthesis by *F. oxysporum*.

**Figure 5 biomolecules-14-01170-f005:**
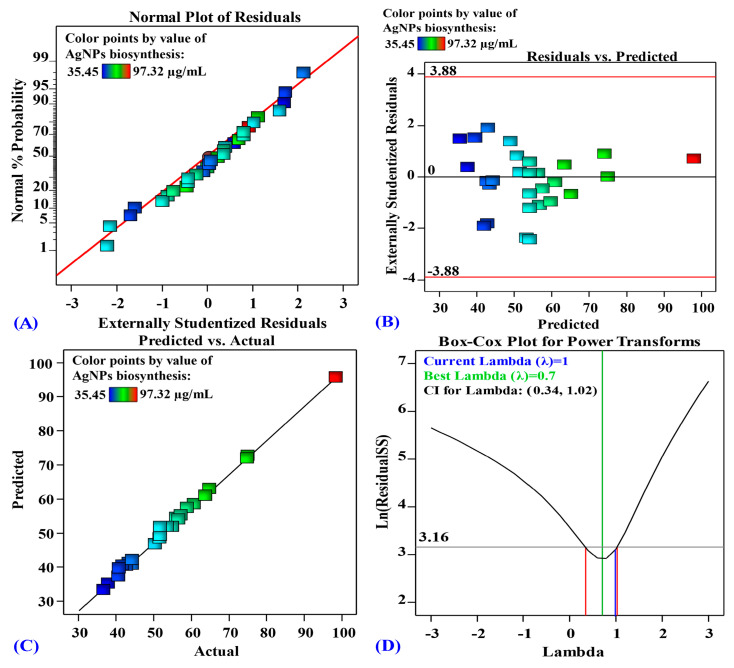
(**A**) Normal probability plot of internally studentized residuals; (**B**) plot of internally studentized residuals versus predicted values, (**C**) plot of predicted versus actual; and (**D**) Box-Cox plot of model transformation of silver nanoparticle biosynthesis by *F. oxysporum* as affected by incubation time (X_1_), initial pH level (X_2_), AgNO_3_ conc. (X_3_), and temperature (X_4_).

**Figure 6 biomolecules-14-01170-f006:**
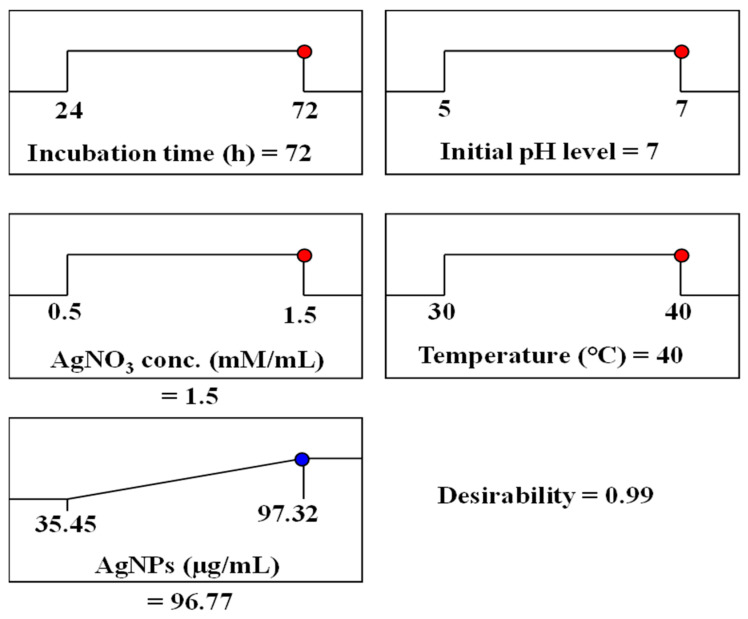
The optimization plot displayed the desirability function and the optimum predicted values of silver nanoparticle biosynthesis by *F. oxysporum*.

**Figure 7 biomolecules-14-01170-f007:**
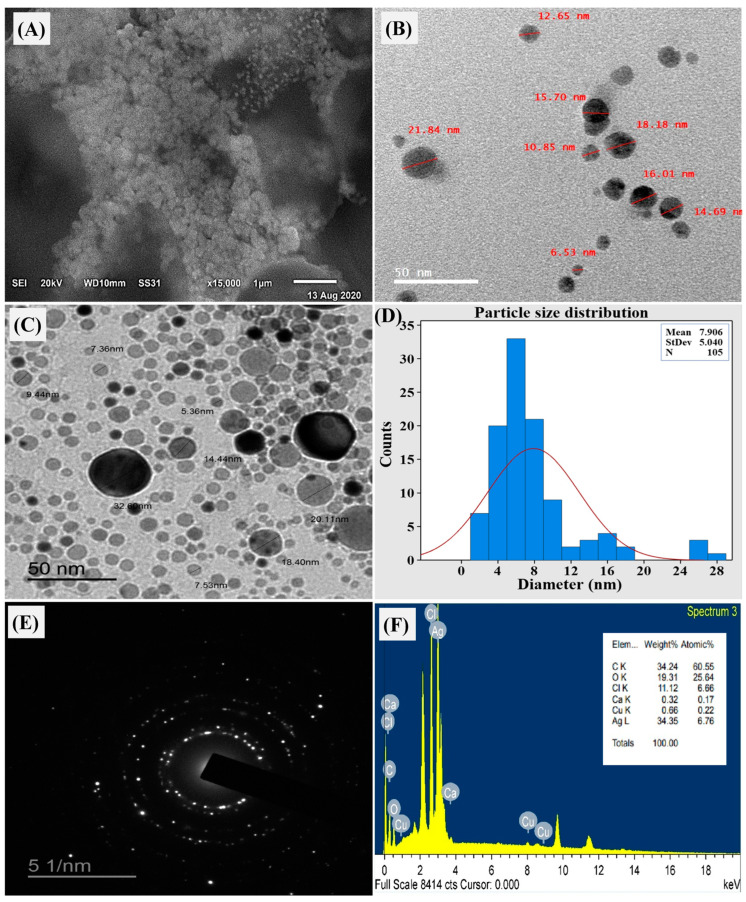
Biosynthesized AgNPs by *F. oxysporum,* where (**A**) scanning electron microscopy (SEM); (**B**,**C**) TEM micrograph; (**D**) particle size distribution; (**E**) selected area diffraction pattern for one nanosilver particle; and (**F**) EDX analysis showing the elemental component of native silver.

**Figure 8 biomolecules-14-01170-f008:**
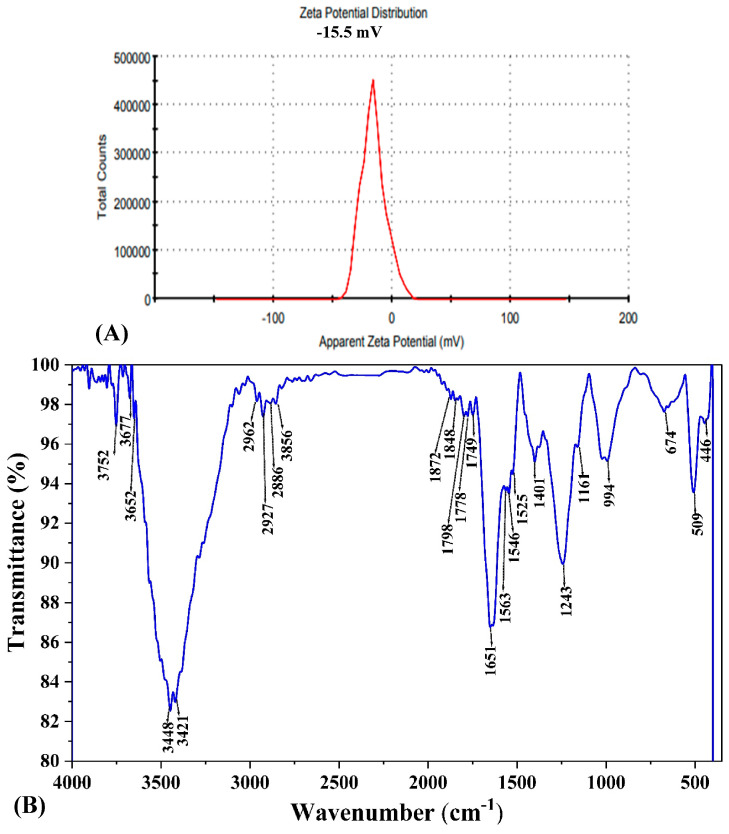
AgNPs synthesized by *F. oxysporum* showing (**A**) Zeta potential and (**B**) Fourier transform infrared (FTIR) spectrum of different functional groups responsible to stabilize or cap AgNPs.

**Figure 9 biomolecules-14-01170-f009:**
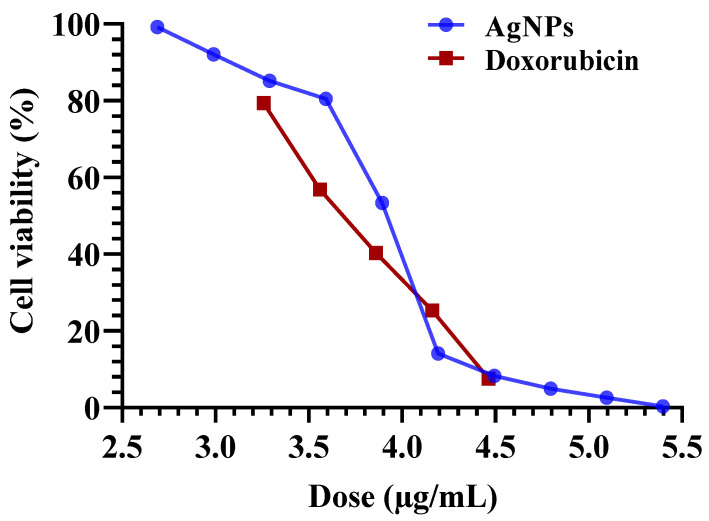
Concentration-response plot of the cytotoxicity of AgNPs biosynthesized using *F. oxysporum* aqueous mycelial-free filtrate with Doxo (positive control) against hepatocellular cancer cells (HepG-2).

**Figure 10 biomolecules-14-01170-f010:**
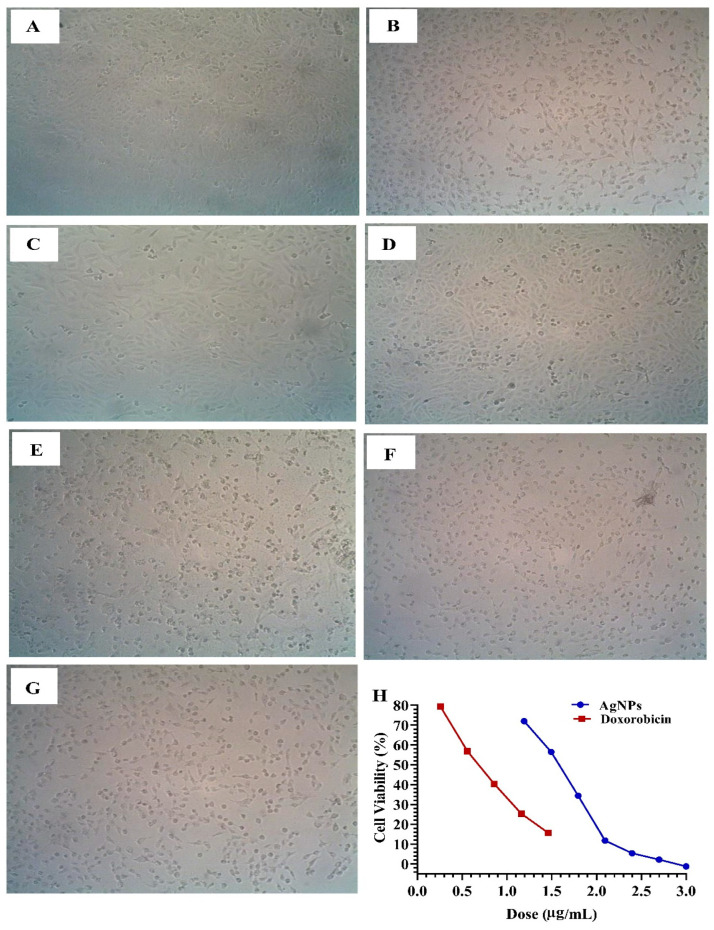
Cytotoxicity of AgNPs biosynthesized using *F. oxysporum* aqueous mycelial-free filtrate showing morphological changes against breast cancer cells (MCF-7). (**A**) living cells (control), (**B**) 1 µg of AgNPs, (**C**) 15.62 µg of AgNPs, (**D**) 31.2 µg of AgNPs, (**E**) 62.5 µg of AgNPs, (**F**) 125 µg of AgNPs, (**G**) 250 µg of AgNPs, and (**H**) concentration–response plot of the cytotoxicity of AgNPs with Doxo (positive control) against breast cancer cells (MCF-7) cell line. From (**A**–**G**) the magnification power at 100×.

**Figure 11 biomolecules-14-01170-f011:**
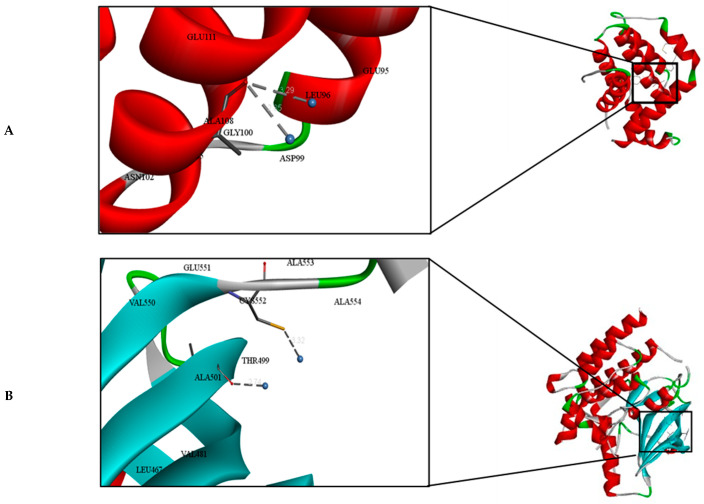
Molecular docking and interactions between silver atoms and the amino acid residues of (**A**) Bcl-2 and (**B**) FGF19 proteins.

**Table 1 biomolecules-14-01170-t001:** Face-centered central composite design (FCCD) matrix representing silver nanoparticle biosynthesis by *Fusarium oxysporum* as affected by incubation time (X_1_), initial pH level (X_2_), AgNO_3_ conc. (X_3_), and temperature (X_4_) with coded and actual factor levels.

Std	Run	Variables				The Yield (%)	AgNPs (µg/mL)		Residuals
		X_1_	X_2_	X_3_	X_4_		Experimental	Predicted	
27	1	0	0	0	0	31.33	53.22	52.89	0.33
29	2	0	0	0	0	30.77	52.27	52.89	−0.62
5	3	−1	−1	1	−1	15.49	39.46	38.34	1.11
15	4	−1	1	1	1	28.9	73.65	72.91	0.74
10	5	1	−1	−1	1	50.67	43.04	41.71	1.33
9	6	−1	−1	−1	1	49.57	42.1	42.23	−0.13
11	7	−1	1	−1	1	48.86	41.5	41.54	−0.04
18	8	1	0	0	0	33.98	57.73	58.48	−0.75
16	9	1	1	1	1	38.19	97.32	96.72	0.6
28	10	0	0	0	0	31.64	53.74	52.89	0.85
21	11	0	0	−1	0	50.73	43.09	43.11	−0.02
1	12	−1	−1	−1	−1	64.52	54.8	55.5	−0.70
26	13	0	0	0	0	31.64	53.74	52.89	0.85
3	14	−1	1	−1	−1	43.16	36.66	36.29	0.37
6	15	1	−1	1	−1	23.36	59.51	59.57	−0.07
4	16	1	1	−1	−1	41.74	35.45	34.35	1.1
22	17	0	0	1	0	24.58	62.63	62.09	0.54
12	18	1	1	−1	1	47.62	40.45	41.59	−1.14
7	19	−1	1	1	−1	19.84	50.56	51.99	−1.43
14	20	1	−1	1	1	24.95	63.58	63.98	−0.40
2	21	1	−1	−1	−1	61.47	52.21	52.98	−0.77
20	22	0	1	0	0	32.99	56.04	56.34	−0.30
17	23	−1	0	0	0	28.91	49.11	47.83	1.28
19	24	0	−1	0	0	29.67	50.4	49.57	0.83
13	25	−1	−1	1	1	15.53	39.56	40.76	−1.20
24	26	0	0	0	1	32.6	55.38	55.13	0.25
25	27	0	0	0	0	30.77	52.27	52.89	−0.62
30	28	0	0	0	0	29.73	50.51	52.89	−2.37
23	29	0	0	0	−1	29.78	50.58	50.3	0.28
8	30	1	1	1	−1	29	73.9	73.8	0.1
**Variable**	**Code**	**Coded and Actual Levels**							
		**−1**	**0**	**1**					
Incubation time (h)	X_1_	24	48	72					
Initial pH level	X_2_	5	6	7					
AgNO_3_ conc. (mM/mL)	X_3_	0.5	1	1.5					
Temperature (°C)	X_4_	30	35	40					

**Table 2 biomolecules-14-01170-t002:** Analysis of variance of CCD for silver nanoparticle biosynthesis by *F. oxysporum* as affected by incubation time (X_1_), initial pH level (X_2_), AgNO_3_ conc. (X_3_), and temperature (X_4_).

Source of Variance		Coefficient Estimate	Sum of Squares	Degrees of Freedom	Mean Square	*F*-Value	*p*-Value
Model	Intercept	52.89	4678.97	14	334.21	220.94	<0.0001 *
Linear effect	X_1_	5.32	509.81	1	509.81	337.02	<0.0001 *
	X_2_	3.38	205.85	1	205.85	136.08	<0.0001 *
	X_3_	9.49	1621.85	1	1621.85	1072.17	<0.0001 *
	X_4_	2.41	104.85	1	104.85	69.31	<0.0001 *
Interaction effect	X_1_ X_2_	0.14	0.34	1	0.34	0.22	0.6442
	X_1_ X_3_	5.94	563.90	1	563.90	372.78	<0.0001 *
	X_1_ X_4_	0.50	3.98	1	3.98	2.63	0.1258
	X_2_ X_3_	8.21	1079.34	1	1079.34	713.52	<0.0001 *
	X_2_ X_4_	4.63	342.69	1	342.69	226.54	<0.0001 *
	X_3_ X_4_	3.92	245.91	1	245.91	162.56	<0.0001 *
Quadratic effect	X_1_^2^	0.27	0.19	1	0.19	0.12	0.7301
	X_2_^2^	0.07	0.01	1	0.01	0.01	0.9281
	X_3_^2^	−0.29	0.21	1	0.21	0.14	0.7117
	X_4_^2^	−0.17	0.08	1	0.08	0.05	0.8258
Error effect	Lack of Fit		15.15	10	1.51	1.00	0.5327
	Pure Error		7.54	5	1.51		
R^2^	0.9952						
Adj R^2^	0.9907						
Pred R^2^	0.9745						
Adeq Precision	71.71						

* Significant values, *F*: Fishers’s function, *p*: level of significance.

**Table 3 biomolecules-14-01170-t003:** Fit summary of FCCD for silver nanoparticle biosynthesis by *F. oxysporum* as affected by incubation time (X_1_), initial pH level (X_2_), AgNO_3_ conc. (X_3_), and temperature (X_4_).

Sequential Model Sum of Squares					
**Source**	**Sum of Squares**	** *df* **	**Mean Square**	***F*-Value**	***p*-Value** ***P*rob > *F***
Linear vs. Mean	2442.37	4	610.59	6.76	0.0008 *
2FI vs. Linear	2236.15	6	372.69	305.96	<0.0001 *
Quadratic vs. 2FI	0.45	4	0.11	0.07	0.9888
Lack of Fit Tests					
**Source**	**Sum of Squares**	** *df* **	**Mean Square**	***F*-value**	***p*-value** ***P*rob > *F***
Linear	2251.76	20	112.59	74.65	<0.0001 *
2FI	15.60	14	1.11	0.74	0.7005
Quadratic	15.15	10	1.51	1.00	0.5327
Model Summary Statistics					
**Source**	**Standard deviation**	**R-Squared**	**Adjusted R-Squared**	**Predicted R-Squared**	**PRESS**
Linear	9.51	0.5195	0.4426	0.1346	4068.63
2FI	1.1	0.9951	0.9925	0.9794	96.74
Quadratic	1.23	0.9952	0.9907	0.9745	119.82

* Significant values, *df*: degree of freedom, PRESS: sum of squares of prediction error, two factors interaction: 2FI.

**Table 4 biomolecules-14-01170-t004:** Cytotoxicity activities (IC_50_) of AgNPs against hepatocellular cancer (HepG-2) and breast cancer cells (MCF-7).

Cell Lines	IC_50_ µg/mL	
	AgNPs	Doxorubicin
HepG2	7.6	5.07
MCF-7	35.4	5.24

**Table 5 biomolecules-14-01170-t005:** Molecular docking analysis of silver molecules.

Molecules	Binding Energy (Kcal/mol)	Inhibition Constant ki (mM)	Metal Chelating Residues	Distance (Å)
Bcl-2	−0.55	397.51	Ag-LEU96	3.29
			Ag-ASP99	2.85
FGF19	−0.54	399.00	Ag-CYS552	3.32
			Ag-THR499	2.74
			Ag-ALA501	2.74

## Data Availability

All data generated or analyzed during this study are included in this published article.

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
