# Peer review of "Myco-Biosynthesis of Silver Nanoparticles, Optimization, Characterization, and In Silico Anticancer Activities by Molecular Docking Approach against Hepatic and Breast Cancer"

_biomolecules, 2024, doi:10.3390/biom14091170_

Round 1
Reviewer 1 Report
Comments and Suggestions for Authors
The manuscript with the title "Myco-biosynthesis of silver nanoparticles, optimization, char-acterization and in silico anticancer activities by molecular docking approach against hepatic and breast cancer" present an interesting study of Ag NPs synthesized by biological means. The study is complete, it contains all the necessary experiments and the data are well-correlated. Excepting for some typos and some missing italic characters there are no other objections.
Author Response
Reviewer #1:
- Reviewer comment: The manuscript with the title "Myco-biosynthesis of silver nanoparticles, optimization, characterization and in silico anticancer activities by molecular docking approach against hepatic and breast cancer" present an interesting study of AgNPs synthesized by biological means. The study is complete, it contains all the necessary experiments, and the data are well-correlated. Excepting for some typos and some missing italic characters there are no other objections.
Authors response: Thanks for your positive comments. We would like to extend a sincere thank you to our reviewer for constructive comments on our work. We checked and improved it.
Reviewer 2 Report
Comments and Suggestions for Authors
This study focuses on the green synthesis of silver nanoparticles (AgNPs) using the extracellular filtrate of Fusarium oxysporum as a reducing agent, to explore their potential anticancer properties. A face-centered central composite design (FCCD) optimized the biosynthesis, achieving a high yield of 96.77 µg/mL under optimal conditions. The researchers conducted several physicochemical characterizations to support their findings. The manuscript is well-written, effectively presenting evidence and discussing the results. However, a few minor comments need to be addressed before it can be accepted for publication.
1. In the introduction, the authors have not highlighted the advantages of silver nanoparticles (AgNPs) over other metal nanoparticles, such as iron oxide and gold NPs.
2. In the legend of Figure 3D, it would be beneficial to highlight or state the type of tubes used to represent different trial samples.
3. On page 13, there is an unlabeled table that needs to be corrected accordingly.
4. The legends of Figures 9 and 10 contain a typo (two "g"s are present).
5. In Figure 9B, is the image of 1 µg/mL AgNPs? Surprisingly, the cells appear so dirty at such a low concentration.
6. I wonder why the authors did not conduct dynamic light scattering (DLS) to measure the size of AgNPs, as they did use it to test zeta potential.
Author Response
Reviewer #2:
- Reviewer comment: This study focuses on the green synthesis of silver nanoparticles (AgNPs) using the extracellular filtrate of Fusarium oxysporum as a reducing agent, to explore their potential anticancer properties. A face-centered central composite design (FCCD) optimized the biosynthesis, achieving a high yield of 96.77 µg/mL under optimal conditions. The researchers conducted several physicochemical characterizations to support their findings. The manuscript is well-written, effectively presenting evidence and discussing the results. However, a few minor comments need to be addressed before it can be accepted for publication.
- Reviewer comment: In the introduction, the authors have not highlighted the advantages of silver nanoparticles (AgNPs) over other metal nanoparticles, such as iron oxide and gold NPs.
Authors response: Thanks for your comment. We added a paragraph to highlight the silver nanoparticles over other nanoparticles in the introduction section.
The use of Cu, Au, Pt, or Pd NPs for plasmonic applications is explored. Ag NPs offer the best quality factor in plasmonic ability and exhibit the plasmonic band in an extensive range of wavelengths (from near ultraviolet to near infrared spectrum), unlike other NPs mentioned. Furthermore, Au and Pt NPs are almost 100 times more expensive than Ag NPs, but Cu NPs utilisation is significantly more problematic due to their high susceptibility to oxidation and the scarcity of recognised nanostructures (Rycenga et al., 2011; Pryshchepa et al., 2020).
Rycenga M, Cobley CM, Zeng J, Li W, Moran CH, Zhang Q, et al. Controlling the synthesis and assembly of silver nanostructures for plasmonic applications. Chem Rev
2011;111: 3669–712. https://doi.org/10.1021/cr100275d.
Pryshchepa O., Pomastowski P., Buszewski B., Silver nanoparticles: Synthesis, investigation techniques, and properties, Advances in Colloid and Interface Science,Volume 284,2020,102246, https://doi.org/10.1016/j.cis.2020.102246.
- Reviewer comment: In the legend of Figure 3D, it would be beneficial to highlight or state the type of tubes used to represent different trial samples.
Authors response: Thanks for your comment. We mentioned each tube on the figure and modified it.
- Reviewer comment: On page 13, there is an unlabeled table that needs to be corrected accordingly.
Authors response: Thanks for your comment. We fixed it.
- Reviewer comment: The legends of Figures 9 and 10 contain a typo (two "g"s are present).
Authors response: Thanks for your comment. We modified the figures legends.
- Reviewer comment: In Figure 9B, is the image of 1 µg/mL AgNPs? Surprisingly, the cells appear so dirty at such a low concentration.
Authors response: Thanks for your comment. We modified Figure 9 completely.
- Reviewer comment: I wonder why the authors did not conduct dynamic light scattering (DLS) to measure the size of AgNPs, as they did use it to test zeta potential.
Authors response: Thanks for your comment. The sizes recorded by particle size analyzer (dynamic light scattering, DLS) can only be taken as a relative value and cannot be compared with that determined by electron microscopy (Gong, et al., 2017). Electron microscopy capable of detecting geometric dimensions of the particles given by measuring the width of individual particles from the image, and determining their shape and surface structure (e.g. texture) (Amini et al., 2016; Kim et al., 2019). Imaging was favored because of its high-resolution visualization of particles and the minimal effect of artifacts on size determination (Kim et al., 2019).
Gong, J., Li, J., Xu, J., Xiang, Z., & Mo, L. Research on cellulose nanocrystals produced from cellulose sources with various polymorphs. RSC advances 7(53), 33486-33493 (2017).
Amini, R., Brar, S. K., Cledon, M. & Surampalli, R. Y. Intertechnique comparisons for nanoparticle size measurements and shape distribution. J Hazard Toxic Radioact Waste 20, B4015004, (2016).
Kim, A., Ng, W. B., Bernt, W., & Cho, N. J. Validation of size estimation of nanoparticle tracking analysis on polydisperse macromolecule assembly. Scientific Reports 9(1), 1-14 (2019).
Reviewer 3 Report
Comments and Suggestions for Authors
In my opinion, the manuscript entitled “Myco-biosynthesis of Silver Nanoparticles: Optimization, Characterization, and In Silico Anticancer Activities by Molecular Docking Approach Against Hepatic and Breast Cancer” presents interesting studies. The authors used a variety of experimental techniques and theoretical approaches to prepare this work. Nevertheless, the use of diverse instruments does not necessarily reflect the quality of the work, and the authors should improve this study in several areas.
- The quality of the figures is quite poor. For example, in Fig. 9, the changes in cell morphology are not visible due to low resolution. Additionally, the images of the suspensions are unnecessary.
- The lack of determination of AgNP concentration in the suspension using an experimental technique is a significant disadvantage of the work. While the authors assess the cytotoxic effect of the nanoparticles, they did not describe the experimental method used to determine a key parameter: the concentration of silver after synthesis. They reported suspending the nanoparticles in solution but did not describe the synthesis method adequately.
- The value provided to describe the yield of the reaction is not correct. In my opinion, the yield of the synthesis should be expressed as a percentage. By knowing the initial concentration of silver and the concentration of silver in the form of nanoparticles after synthesis (after removing any ionic silver), the efficiency should be determined. The authors did not provide the concentration of silver in nanoparticle form or specify whether they purified the suspension to remove silver in ionic form or only the complexes formed during synthesis.
- The authors did not conducted analysis of zeta potential. The value without the determination of pH, ionic strenght of suspension does not make sens. The zeta potential depends on these parameteres. Morover, they should described the model used for determination of this parameter.
- Optimization of Biosynthesis Parameters using Face-Centered Central Composite De- 187 sign (FCCD) – in this field, it is important to explain why the amount of Fungal biomass was not considered.
- how about the toxicity of fungal biomass?
- When analyzing the morphology of AgNPs, a size distribution should be provided. The values described in the results differ from those mentioned in the conclusions.
- The authors should explain why the particles synthesized by them are superior to those obtained using other chemical methods, particularly in terms of biological activity.
I am not able to suport publication of this report without the detection of mass concentration of silver nanoparticles in the suspensions used in the biological tests.
Author Response
Reviewer #3:
- Reviewer comment: In my opinion, the manuscript entitled “Myco-biosynthesis of Silver Nanoparticles: Optimization, Characterization, and In Silico Anticancer Activities by Molecular Docking Approach Against Hepatic and Breast Cancer” presents interesting studies. The authors used a variety of experimental techniques and theoretical approaches to prepare this work. Nevertheless, the use of diverse instruments does not necessarily reflect the quality of the work, and the authors should improve this study in several areas.
Authors response: We would like to send a sincere thanks to our reviewer for thoughtful critiques of our manuscript. We agreed with all comments and suggestions of our reviewer. We found them quite useful as we improve our paper.
- Reviewer comment: The quality of the figures is quite poor. For example, in Fig. 9, the changes in cell morphology are not visible due to low resolution. Additionally, the images of the suspensions are unnecessary.
Authors response: Agree. Per your suggestion, Fig. 9, the changes in cell morphology have been omitted due to low resolution. Additionally, the images of the suspensions have been ommitted.
- Reviewer comment: The lack of determination of AgNP concentration in the suspension using an experimental technique is a significant disadvantage of the work. While the authors assess the cytotoxic effect of the nanoparticles, they did not describe the experimental method used to determine a key parameter: the concentration of silver after synthesis. They reported suspending the nanoparticles in solution but did not describe the synthesis method adequately.
Authors response: Agree. The reviewer is right, and the green synthesis of AgNPs, determination of AgNPs concentration in the suspension using an experimental technique and all the requested information has been added in the revised version as follows:
Green Synthesis of AgNPs:
In a 250 mL Erlenmeyer flask, 100 mL of the aqueous mycelial free filtrate of F. oxysporum was treated with precisely weighed AgNO3 to reach a final concentration of 1 mM AgNO3 and incubated in an orbital shaker with continuous agitation (150 rpm) under dark conditions (to prevent the photooxidation of silver ions) for 72 h at 30°C. The aqueous mycelial free filtrate without addition of AgNO3 was maintained as a control (Devi and Joshi, 2012). Following incubation, the mixture of AgNO3 and the aqueous mycelial free filtrate of F. oxysporum turned to dark brown, the visual color change indicates the formation of AgNPs. On contrast no color change was noticed with the absence of AgNO3. The maximum absorbance of the green synthesized AgNPs was scanned in the wavelength range of 300–600 nm range by UV-Vis spectroscopy.
Ultraviolet-visible (UV-Vis) Spectroscopy
Atomic absorption spectroscopy (AAS) is used to determine the concentration of AgNPs, however it does not reflect the concentrations of AgNPs particles themselves. Instead, it reflects the concentration of Ag+ ions. Therefore, unlike the AAS spectrum, if UV-Vis spectra were used, the concentration of AgNPs would not include Ag+ ions but for only nanoparticles. AgNPs prepared by using the aqueous mycelial free filtrate of Fusarium oxysporum. Only AgNPs have UV-Vis spectra at wavelengths from 400 nm to 450 nm depending on their size and shape; however, the height of absorption intensity value will rely on the quantity of AgNPs in the solution. The concentration of AgNPs can be quantitatively determined based on the linear dependence between AgNPs concentration and UV-Vis absorbance up to 200 ppm (Hung et al., 2018). To quantify AgNPs using the UV-Vis method, a calibration curve must be generated from the absorption intensity values of the UV-Vis spectrum determined from a known concentration solution. The variations in the UV-Vis spectra are indicative of the AgNPs' particle size and the solution's color. Different AgNPs concentration series (10–100 μg/mL) was obtained by diluting AgNPs stock solution (500 μg/mL) with deionized water. Those standards were measured together with samples to obtain the corresponding UV-Vis and the calibration curve for calculating sample concentrations.
Devi, L. S., & Joshi, S. R. (2012). Antimicrobial and synergistic effects of silver nanoparticles synthesized using soil fungi of high altitudes of eastern Himalaya. Mycobiology, 40(1), 27-34.
Hung, N. D., Nam, V. N., ThiNhan, T., & Dung, T. T. N. (2018). Quantitative concentration determination of silver nanoparticles prepared by DC high voltage electrochemical method. Vietnam Journal of Chemistry, 56(5), 553-558.
- Reviewer comment: The value provided to describe the yield of the reaction is not correct. In my opinion, the yield of the synthesis should be expressed as a percentage. By knowing the initial concentration of silver and the concentration of silver in the form of nanoparticles after synthesis (after removing any ionic silver), the efficiency should be determined. The authors did not provide the concentration of silver in nanoparticle form or specify whether they purified the suspension to remove silver in ionic form or only the complexes formed during synthesis.
Authors response: Agree. As suggested by the reviewer, the yield (%) has been calculated and added to Table 1 as the following:
Table 1. Face-centered central composite design (FCCD) matrix representing silver nanoparticles biosynthesis by Fusarium oxysporum as affected by incubation time (X1), initial pH level (X2), AgNO3 conc. (X3) and temperature (X4) with coded and actual factor levels.
Std |
Run |
Variables |
The yield (%) |
AgNPs (µg/mL) |
Residuals |
||||
X1 |
X2 |
X3 |
X4 |
Experimental |
Predicted |
||||
27 |
1 |
0 |
0 |
0 |
0 |
31.33 |
53.22 |
52.89 |
0.33 |
29 |
2 |
0 |
0 |
0 |
0 |
30.77 |
52.27 |
52.89 |
-0.62 |
5 |
3 |
-1 |
-1 |
1 |
-1 |
15.49 |
39.46 |
38.34 |
1.11 |
15 |
4 |
-1 |
1 |
1 |
1 |
28.90 |
73.65 |
72.91 |
0.74 |
10 |
5 |
1 |
-1 |
-1 |
1 |
50.67 |
43.04 |
41.71 |
1.33 |
9 |
6 |
-1 |
-1 |
-1 |
1 |
49.57 |
42.10 |
42.23 |
-0.13 |
11 |
7 |
-1 |
1 |
-1 |
1 |
48.86 |
41.50 |
41.54 |
-0.04 |
18 |
8 |
1 |
0 |
0 |
0 |
33.98 |
57.73 |
58.48 |
-0.75 |
16 |
9 |
1 |
1 |
1 |
1 |
38.19 |
97.32 |
96.72 |
0.60 |
28 |
10 |
0 |
0 |
0 |
0 |
31.64 |
53.74 |
52.89 |
0.85 |
21 |
11 |
0 |
0 |
-1 |
0 |
50.73 |
43.09 |
43.11 |
-0.02 |
1 |
12 |
-1 |
-1 |
-1 |
-1 |
64.52 |
54.80 |
55.50 |
-0.70 |
26 |
13 |
0 |
0 |
0 |
0 |
31.64 |
53.74 |
52.89 |
0.85 |
3 |
14 |
-1 |
1 |
-1 |
-1 |
43.16 |
36.66 |
36.29 |
0.37 |
6 |
15 |
1 |
-1 |
1 |
-1 |
23.36 |
59.51 |
59.57 |
-0.07 |
4 |
16 |
1 |
1 |
-1 |
-1 |
41.74 |
35.45 |
34.35 |
1.10 |
22 |
17 |
0 |
0 |
1 |
0 |
24.58 |
62.63 |
62.09 |
0.54 |
12 |
18 |
1 |
1 |
-1 |
1 |
47.62 |
40.45 |
41.59 |
-1.14 |
7 |
19 |
-1 |
1 |
1 |
-1 |
19.84 |
50.56 |
51.99 |
-1.43 |
14 |
20 |
1 |
-1 |
1 |
1 |
24.95 |
63.58 |
63.98 |
-0.40 |
2 |
21 |
1 |
-1 |
-1 |
-1 |
61.47 |
52.21 |
52.98 |
-0.77 |
20 |
22 |
0 |
1 |
0 |
0 |
32.99 |
56.04 |
56.34 |
-0.30 |
17 |
23 |
-1 |
0 |
0 |
0 |
28.91 |
49.11 |
47.83 |
1.28 |
19 |
24 |
0 |
-1 |
0 |
0 |
29.67 |
50.40 |
49.57 |
0.83 |
13 |
25 |
-1 |
-1 |
1 |
1 |
15.53 |
39.56 |
40.76 |
-1.20 |
24 |
26 |
0 |
0 |
0 |
1 |
32.60 |
55.38 |
55.13 |
0.25 |
25 |
27 |
0 |
0 |
0 |
0 |
30.77 |
52.27 |
52.89 |
-0.62 |
30 |
28 |
0 |
0 |
0 |
0 |
29.73 |
50.51 |
52.89 |
-2.37 |
23 |
29 |
0 |
0 |
0 |
-1 |
29.78 |
50.58 |
50.30 |
0.28 |
8 |
30 |
1 |
1 |
1 |
-1 |
29.00 |
73.90 |
73.80 |
0.10 |
Variable |
Code |
Coded and actual levels |
||
-1 |
0 |
1 |
||
Incubation time (h) |
X1 |
24 |
48 |
72 |
Initial pH level |
X2 |
5 |
6 |
7 |
AgNO3 conc. (mM/mL) |
X3 |
0.5 |
1 |
1.5 |
Temperature (°C) |
X4 |
30 |
35 |
40 |
- Reviewer comment: how about the toxicity of fungal biomass?
Authors response: Agree. As suggested by the reviewer, the explanation of the cytotoxicity of nanoparticles from Fusarium species has been added in the revised version as follows:
Rai et al. (2021) reported that the cytotoxicity of nanoparticles from Fusarium species, in particular AgNPs, and their underlying molecular mechanisms have been demonstrated in numerous studies that have applied a variety of model cell lines. The biogenic nanoparticles are capped with a corona, which is composed of natural molecules, including proteins. This nanoparticle corona has a substantial impact on the biological response. The corona can be classified into two types: hard corona and soft corona, based on the surface affinity as well as exchange rate. While the rigid coronas are rigid for entry into the cellular system, the soft corona proteins function as "vehicles" for the silver ions. Several reports demonstrate that the nanoparticle corona's proteins interact with the cells, rather than the nanoparticles themselves. For this reason, the corona formation and composition have major effects for both internalization and toxicity (Lesniak et al., 2012). In addition to the protein charges, these functional groups also modulate the cytotoxic properties of the nanoparticle corona. The biological activity of nanoparticles is significantly influenced by their surface charge. In summary, biological fluid properties, composition, size, shape, and surface characteristics of particles all influence the corona structure and, thus, their detrimental effects on the environment and the health of humans (Rai et al., 2021).
Rai, M., Bonde, S., Golinska, P., Trzcińska-Wencel, J., Gade, A., Abd-Elsalam, K. A., ... & Ingle, A. P. (2021). Fusarium as a novel fungus for the synthesis of nanoparticles: mechanism and applications. Journal of Fungi, 7(2), 139.
Lesniak, A., Fenaroli, F., Monopoli, M. P., Åberg, C., Dawson, K. A., & Salvati, A. (2012). Effects of the presence or absence of a protein corona on silica nanoparticle uptake and impact on cells. ACS nano, 6(7), 5845-5857.
- Reviewer comment: When analyzing the morphology of AgNPs, a size distribution should be provided. The values described in the results differ from those mentioned in the conclusions.
Author response: Agree. As suggested by the reviewer, the size distribution has been included in Figure 7 in the revised version.
Figure 7. Biosynthesized AgNPs by F. oxysporum where; A) scanning electron microscopy (SEM); B, C) TEM micrograph; D) Particle size distribution; E) Selected area diffraction pattern for one nano-silver particle and F) EDX analysis showing the elemental component of native silver.
- Reviewer comment: The authors did not conducted analysis of zeta potential. The value without the determination of pH, ionic strength of suspension does not make sense The zeta potential depends on these parameters. Moreover, they should describe the model used for determination of this parameter.
Author response: Agree. We appreciate this great advice and we totally agree with you, but unfortunately, due to lack of resources and fundamental support, we are unable to do the impact of environmental conditions such as pH, ionic strength on the zeta potential of silver nanoparticles suspensions at the present study. In addition, the available sample size is not sufficient to conduct the recommended analyses. We will take your advice into account and also recommend study of the impact of pH, ionic strength on the zeta potential of silver nanoparticles suspensions to be performed on the same isolate in the upcoming studies.Thank you.
- Reviewer comment: Optimization of Biosynthesis Parameters using Face-Centered Central Composite De- 187 sign (FCCD) – in this field, it is important to explain why the amount of Fungal biomass was not considered.
Author response: Agree. Four variables have been chosen from the others variables to we can manage the experiment in 30 runs; if we choose five variables, the design must be 50 runs. Thus, in a preliminary experiment, we investigate the impact of different amounts of fungal biomass on the synthesis of silver nanoparticles, relying on previous studies. The amount of 10 g/L was optimal.
- Reviewer comment: I am not able to support publication of this report without the detection of mass concentration of silver nanoparticles in the suspensions used in the biological tests.
Author response: Thanks for your comment and I agree with you. The reviewer is correct; sorry for this is a mistake; the text has been corrected in the revised manuscript as the following: (2.7-5.4 μg/mL).
Reviewer 4 Report
Comments and Suggestions for Authors
The article "Myco-biosynthesis of silver nanoparticles, optimization, characterization and in silico anticancer activities by molecular docking approach against hepatic and breast cancer" describes the biosynthesis of silver nanoparticles by microorganisms.
While the article shows promise as a valuable study, it requires some revisions before it can be considered for publication in this journal:
The particles and other structures are considered belonging to nano domain when at least one dimension is under 100 nm.
This work is interesting and can be boosted further. Hence the following literature could prove this manuscript doi: 10.1155/2020/6651207; doi: 10.3390/ijms24065865; doi: 10.3390/ijms231810626
Use proper indices for formula like AgNO3 (row 632).
Abbreviations should be expanded upon first mention in the text (e.g. d H2O).
The English language needs some polishing for style and typos (e.g. row 635 “invitro” missing space). Additionally, all Latin names/expressions should be italicized.
Please format consistently the section sub-titles.
For “Selected area diffraction pattern” the wide used name is Selected area electron diffraction (SAED) – please rephrase row 248 and section 3.8.1.
EDX is a technique that identifies elements, therefore should be chlorine not chloride.
As authors determined the antitumoral activity, they should propose a way for in vivo treatment.
Author Response
- Reviewer comment: The article "Myco-biosynthesis of silver nanoparticles, optimization, characterization and in silico anticancer activities by molecular docking approach against hepatic and breast cancer" describes the biosynthesis of silver nanoparticles by microorganisms. While the article shows promise as a valuable study, it requires some revisions before it can be considered for publication in this journal: The particles and other structures are considered belonging to nano domain when at least one dimension is under 100 nm.
Authors response: Thanks for your comment. We made all required corrections.
- Reviewer comment: This work is interesting and can be boosted further. Hence the following literature could prove this manuscript doi: 10.1155/2020/6651207; doi: 10.3390/ijms24065865; doi: 10.3390/ijms231810626
Authors response: Thanks for your comment. We added the suggested references:
- Marinescu L, Ficai D, Oprea O, Marin A, Ficai A, Andronescu E, Holban A-M, and Soare LC. 2020. Optimized Synthesis Approaches of Metal Nanoparticles with Antimicrobial Applications. J. Nanomaterials 2020 (2020). https://doi.org/10.1155/2020/6651207.
- Zeng YJ, Wu XL, Yang HR, Zong MH, Lou WY. 1,4-α-Glucosidase from Fusarium solanifor Controllable Biosynthesis of Silver Nanoparticles and Their Multifunctional Applications. Int J Mol Sci. 2023;24(6):5865. doi: 10.3390/ijms24065865.
- Mohanta YK, Nayak D, Mishra AK, Chakrabartty I, Ray MK, Mohanta TK, Tayung K, Rajaganesh R, Vasanthakumaran M, Muthupandian S, Murugan K, Sharma G, Dahms HU, Hwang JS. Green Synthesis of Endolichenic Fungi Functionalized Silver Nanoparticles: The Role in Antimicrobial, Anti-Cancer, and Mosquitocidal Activities. Int J Mol Sci. 2022;23(18):10626. doi: 10.3390/ijms231810626.
- Reviewer comment: Use proper indices for formula like AgNO3 (row 632).
Authors response: Thanks for your comment. We corrected it.
- Reviewer comment: Abbreviations should be expanded upon first mention in the text (e.g. d H2O).
Authors response: Thanks for your comment. We revised it.
- Reviewer comment: The English language needs some polishing for style and typos (e.g. row 635 “invitro” missing space). Additionally, all Latin names/expressions should be italicized.
Authors response: Thanks for your comments. We checked the English language of all text and corrected the style and typos. All names were italicized.
- Reviewer comment: Please format consistently the section sub-titles.
Authors response: Thanks for your comment. We modified it.
- Reviewer comment: For “Selected area diffraction pattern” the wide used name is Selected area electron diffraction (SAED) – please rephrase row 248 and section 3.8.1.
Authors response: Thanks for your comment. We changed it.
- Reviewer comment: EDX is a technique that identifies elements, therefore should be chlorine not chloride.
Authors response: Thanks for your comment. We replaced it.
- Reviewer comment: As authors determined the antitumoral activity, they should propose a way for in vivo treatment.
Authors response: Thanks for your comment. We focused on in vitro and in silico studies as it influenced on cellular and molecular targets that control apoptosis.
Thank you very much for your insightful remarks and comments that helped us enhance the manuscript.
Round 2
Reviewer 3 Report
Comments and Suggestions for Authors
The authors considered each of my suggestions and provided precise answers. I also noticed that the manuscript was carefully improved. I really appreciate every improvement and the effort made by the authors. However, I would like to address the determination of the concentration of AgNPs, as I completely disagree with the authors' response on this matter. Please allow me to present my point of view.
First of all, the spectra of AgNPs recorded in the UV-Vis region can exhibit more than one characteristic absorption peak arising from localized surface plasmon resonance (LSPR). For instance, anisotropic AgNPs (e.g., nanowires, nanoprisms) exhibit two or three peaks. In this case, determining the mass concentration of AgNPs in suspension using the authors' approach is impossible.
Secondly, the position of the peak arising from LSPR depends on the surrounding medium, including stabilizing agent molecules. The peak width is correlated with AgNP polydispersity and should not be compared with more monodisperse particles. In your UV-Vis spectrum, one can observe that the absorption values are shifted towards higher levels, likely due to the absorption of organic matter across the measured range. In my opinion, the applied technique for determining the concentration is not appropriate and the determined concentration other than real. Please discuss.
To accurately determine the concentration of AgNPs using AAS, the nanoparticles should be digested in hot, concentrated nitric acid, and after proper sample preparation, the total silver concentration in the sample should be measured. Moreover, it is often necessary to determine the oxidation state of AgNPs and the amount of leached or unreacted silver ions. In this case, before AAS measurements, silver ions and silver nanoparticles should be separated using appropriate membranes. Then, with two samples, one can determine the concentration of ionic silver and silver in the form of AgNPs in the suspension. I encourage you to conduct such an investigation. Furthermore, I think the quality of the figures should be improved. The drawings are still stretched and shapeless. For the zeta potential studies, please add information about the pH of the measured sample and indicate the model used for determining the zeta potential (e.g., Oshima, Smoluchowski, etc.).
Author Response
Reviewer Comment: The authors considered each of my suggestions and provided precise answers. I also noticed that the manuscript was carefully improved. I really appreciate every improvement, and the effort made by the authors. However, I would like to address the determination of the concentration of AgNPs, as I completely disagree with the authors' response on this matter. Please allow me to present my point of view.
Authors Response: Thanks for your supportive comments. We appreciate the reviewer for agreeing with us on the importance and novelty of our study, as well as the important comments. We revised our manuscript according to the comments of the reviewers. The followings are point-by-point responses to your concerns.
Reviewer Comment: First of all, the spectra of AgNPs recorded in the UV-Vis region can exhibit more than one characteristic absorption peak arising from localized surface plasmon resonance (LSPR). For instance, anisotropic AgNPs (e.g., nanowires, nanoprisms) exhibit two or three peaks. In this case, determining the mass concentration of AgNPs in suspension using the authors' approach is impossible.
Authors Response: According to Mie-theory (Mie, 1908), only a single SPR band is expected in the absorption spectra of spherical nanoparticles, whereas anisotropic particles could give rise to two or more (SPR) bands depending on the shape of the particles. The number of SPR peaks increases as the symmetry of the nanoparticle decreases (Sosa et al., 2003).
In the present investigation, the optical characteristics of AgNPs revealed a single peak, and the highest absorbance was measured at a wavelength of 418.3 nm revealing spherical shape of AgNPs, which was further confirmed by TEM images.
Reviewer Comment: Secondly, the position of the peak arising from LSPR depends on the surrounding medium, including stabilizing agent molecules. The peak width is correlated with AgNP polydispersity and should not be compared with more monodisperse particles. In your UV-Vis spectrum, one can observe that the absorption values are shifted towards higher levels, likely due to the absorption of organic matter across the measured range. In my opinion, the applied technique for determining the concentration is not appropriate and the determined concentration other than real. Please discuss.
Authors Response: The SPR peak of silver nanoparticles in aqueous solution shifts to longer wavelengths with increasing particle size. The position and shape of plasmon absorption of silver nanoclusters are strongly dependent on particle size, stabilizing molecules or surface adsorbed particles, as well as the dielectric constant of the medium (Krishnaraj et al., 2010).
The Ag-NPs solution obtained was purified by centrifugation at 5000 rpm. The pellet containing silver nanoparticles was washed 3 – 4 times with deionized water to remove silver ions and the aqueous mycelial free filtrate of F. oxysporum residues. To completely remove the unreacted Ag+ from nanoparticles sample, it was treated with NaCl solution. After addition of NaCl, Ag+ reacts with Cl- and form white precipitate of AgCl. The precipitate was then removed by centrifugation of mixture at 4,000 rpm for 15 minutes. After removing the pellet, the supernatant was centrifuged at 11,000 rpm for 30 min to concentrate the silver nanoparticles, and then dried at 50 °C. The dried biogenic synthesized AgNPs were purified through washing with organic solvent (acetone) and centrifugation. The dried acetone washed AgNPs were used for further studies (Zaki et al., 2019). Subsequently, their dry mass was estimated. To be used in further studies silver bionanoparticles were resuspended in deionized water, to obtain their desired concentration (Wrótniak-Drzewiecka et al., 2014; Ingle et al., 2009).
Wrótniak-Drzewiecka, W., Gaikwad, S., Laskowski, D., Dahm, H., NiedojadÅ‚o, J., Gade, A., & Rai, M. (2014). Novel approach towards synthesis of silver nanoparticles from Myxococcus virescens and their lethality on pathogenic bacterial cells. Austin J Biotechnol Bioeng, 1(1), 7.
Ingle A, Rai M, Gade A, Bawaskar M. Fusarium solani: a novel biological agent for the extracellular synthesis of silver nanoparticles. J. Nanopart. Res. 2009; 11: 2079-2085.
Zaki, Almaz, Md Nafe Aziz, Rakhshan Ahmad, Irshad Ahamad, M. Shadab Ali, Durdana Yasin, Bushra Afzal et al. "Synthesis, purification and characterization of Plectonema derived AgNPs with elucidation of the role of protein in nanoparticle stabilization." RSC advances 12, no. 4 (2022): 2497-2510.
Reviewer Comment: To accurately determine the concentration of AgNPs using AAS, the nanoparticles should be digested in hot, concentrated nitric acid, and after proper sample preparation, the total silver concentration in the sample should be measured. Moreover, it is often necessary to determine the oxidation state of AgNPs and the amount of leached or unreacted silver ions. In this case, before AAS measurements, silver ions and silver nanoparticles should be separated using appropriate membranes. Then, with two samples, one can determine the concentration of ionic silver and silver in the form of AgNPs in the suspension. I encourage you to conduct such an investigation.
Authors Response: The Ag-NPs solution obtained was purified by centrifugation at 5000 rpm. The pellet containing silver nanoparticles was washed 3 – 4 times with deionized water to remove silver ions and the aqueous mycelial free filtrate of F. oxysporum residues. To completely remove the unreacted Ag+ from nanoparticles sample, it was treated with NaCl solution. After addition of NaCl, Ag+ reacts with Cl- and form white precipitate of AgCl. The precipitate was then removed by centrifugation of mixture at 4,000 rpm for 15 minutes. After removing the pellet, the supernatant was centrifuged at 11,000 rpm for 30 min to concentrate the silver nanoparticles, and then dried at 50°C. The dried biogenic synthesized AgNPs were purified through washing with organic solvent (acetone) and centrifugation. The dried acetone washed AgNPs were used for further studies (Zaki et al., 2019). Subsequently, their dry mass was estimated. To be used in further studies silver bionanoparticles were resuspended in deionized water, to obtain their desired concentration (Wrótniak-Drzewiecka et al., 2014; Ingle et al., 2009).
Reviewer Comment: Furthermore, I think the quality of the figures should be improved. The drawings are still stretched and shapeless.
Authors Response: Thanks for your suggestion. The quality of the figures has been improved
Reviewer Comment: For the zeta potential studies, please add information about the pH of the measured sample and indicate the model used for determining the zeta potential (e.g., Oshima, Smoluchowski, etc.).
Authors Response: Thanks for indicating this. We have tried our best to provide this required information by contacting to the centre that performing the zeta potential. We did not follow any of the theories given here. We only used zeta potential measurement as one of the characterisation approaches and did not delve too in-depth with them. Here's the zeta potential report we have.
Thank you very much for your insightful remarks and comments that helped us enhance the manuscript.
